# Cortical bone maturation in mice requires SOCS3 suppression of gp130/STAT3 signalling in osteocytes

Emma C Walker[1], Kim Truong[1,2], Narelle E McGregor[1], Ingrid J Poulton[1], Tsuyoshi Isojima[1,3], Jonathan H Gooi[1,4], T John Martin[1,2], Natalie A Sims[1,2]*

[1]St. Vincent's Institute of Medical Research, Fitzroy, Australia; [2]University of Melbourne, Department of Medicine at St. Vincent's Hospital, Fitzroy, Australia; [3]Department of Pediatrics, Teikyo University School of Medicine, Tokyo, Japan; [4]Department of Biochemistry and Molecular Biology, Bio21 Molecular Science and Biotechnology Institute, University of Melbourne, Parkville, Australia

**Abstract** Bone strength is determined by its dense cortical shell, generated by unknown mechanisms. Here we use the $Dmp1^{Cre}:Socs3^{f/f}$ mouse, with delayed cortical bone consolidation, to characterise cortical maturation and identify control signals. We show that cortical maturation requires a reduction in cortical porosity, and a transition from low to high density bone, which continues even after cortical shape is established. Both processes were delayed in $Dmp1^{Cre}:Socs3^{f/f}$ mice. SOCS3 (suppressor of cytokine signalling 3) inhibits signalling by leptin, G-CSF, and IL-6 family cytokines (gp130). In $Dmp1^{Cre}:Socs3^{f/f}$ bone, STAT3 phosphorylation was prolonged in response to gp130-signalling cytokines, but not G-CSF or leptin. Deletion of gp130 in $Dmp1^{Cre}:Socs3^{f/f}$ mice suppressed STAT3 phosphorylation in osteocytes and osteoclastic resorption within cortical bone, leading to rescue of the corticalisation defect, and restoration of compromised bone strength. We conclude that cortical bone development includes both pore closure and accumulation of high density bone, and that these processes require suppression of gp130-STAT3 signalling in osteocytes.

*For correspondence:
nsims@svi.edu.au

Competing interests: The authors declare that no competing interests exist.

## Introduction

The characteristic external structure of bone (cortical bone) forms during development and continues to consolidate as bones grow in length. During longitudinal growth, thin bony trabeculae arise from the cartilaginous growth plate, and, at the periphery of the metaphysis, they coalesce to form the cortical shell (*Enlow, 1962*). As well as achieving the distinctive thickened cortical shape, the woven bone that makes up the initial cortex is gradually replaced with a stronger layered (lamellar) structure (*Enlow, 1962*). In rodent bone, the woven bone is partially replaced by modelling drift during growth: new lamellar bone is deposited asymmetrically on both inner (endocortical) and outer (periosteal) cortical surfaces, and woven endochondral bone remains in the centre of the cortex (*Shipov et al., 2013*). In larger mammalian bones, including human, in addition to deposition of lamellar bone on endocortical and periosteal surfaces (*Maggiano et al., 2015*), the inner cortical bone is remodelled into lamellar Haversian systems surrounding intracortical blood vessels (*Enlow, 1962*). The cellular changes and molecular mechanisms that control the establishment of the initial compact cortical bone structure remain undefined.

Since corticalisation occurs during rapid longitudinal growth, the process has been difficult to quantify. Bone growth in length occurs at the growth plate, and measurement of a cortical region of interest in the developing cortex in animals and humans during bone lengthening is difficult because a fixed position on the bone cannot be reliably identified due to the non-linear pattern of growth.

Differences in positioning of a region of interest by even 1 to 2 mm may alter the analysis of any structural changes in bone (*Seeman and Ghasem-Zadeh, 2016*). Adding to this challenge, multiple scans of children with developing bones would be required to measure the process of corticalisation in humans, which raises safety concerns about radiation exposure effects on growing bone (*Williams and Davies, 2006*; *Brenner and Hall, 2007*).

A focus on the development and consolidation of cortical bone structure will improve our understanding of a process shared by all vertebrates and improve methods used for age estimation in bio-archaeology and forensic anthropology (*Maggiano et al., 2015*). I, if the signalling pathways controlling cortical structure are identified, this could provide methods to improve bone strength in patients prone to fracture. Peak bone mass achieved in early adulthood strongly predicts later fracture risk after skeletal structure begins to decline with age (*Seeman, 2002*). Abnormalities in trabecular formation from the growth plate influence the structure of bone and determines fragility and risk of fracture; healthy girls and pre- and postmenopausal women with thinner or fewer trabeculae exhibit lower cortical area and higher cortical porosity (*Bala et al., 2015*). In older individuals, a reversal of this process (trabecularisation) also occurs on the endocortical surface of bone, leading to a high cortical porosity and increased skeletal fragility (*Zebaze et al., 2010*); individuals over the age of 65 lose up to 50% of cortical bone mass during this process (*Riggs et al., 1981*). Cortical mass and dimensions determine the strength of long bones (e.g. femur, radius, ulna) (*Ammann and Rizzoli, 2003*), and fractures in long bones account for 80% of all osteoporotic fractures (*Zebaze et al., 2010*; *Clarke, 2008*). However, current treatments for osteoporosis are less effective at cortical sites, even though they prevent fractures in vertebral bone that contains a high proportion of trabecular bone (*Clarke, 2008*; *Lindsay et al., 1997*).

We recently identified the first cell communication mechanism required for cortical bone maturation. Cortical bone consolidation was delayed in *Dmp1^Cre^:Socs3^f/f^* mice lacking suppressor of cytokine signalling 3 (SOCS3) in *Dmp1^Cre^* expressing cells (osteocytes and late osteoblasts), particularly females (*Cho et al., 2017*). In addition, deletion of SOCS3 in the osteo-chondral lineage also delayed formation of dense cortical bone (*Liu et al., 2019*). This indicates that inhibition of cytokine signalling in osteocytes by SOCS3 is needed for timely formation of cortical bone. However, SOCS3 provides negative feedback for a range of cytokine receptors, including the leptin, G-CSF, and gp130 receptors. The latter is utilized by the IL-6 family of cytokines, which includes Interleukin 6 (IL-6), Interleukin 11 (IL-11), oncostatin M (OSM), cardiotrophin 1 (CT-1) and leukaemia inhibitory factor (LIF). Leptin, G-CSF and IL-6 family cytokines all have the potential to modify cortical development since they each promote bone formation through local action in bone (*McGregor et al., 2019*; *Sims et al., 2005*; *Walker et al., 2008*; *Cornish et al., 1993*; *Walker et al., 2010*; *Winkler et al., 2010*; *Scheller et al., 2010*), modify gene expression by osteocytes (*McGregor et al., 2019*; *Walker et al., 2010*), and, in some cases, promote bone resorption (*Tamura et al., 1993*; *Richards et al., 2000*). Although phenotypes caused by SOCS3 deficiency in other organs were rescued by IL-6 deletion (*Croker et al., 2003*), this was not the case in *Dmp1^Cre^:Socs3^f/f^* mice (*Cho et al., 2017*). The specific cytokine receptor that must be suppressed for cortical development remains unidentified.

In our earlier study we realised the limitations of morphological analyses of cortical bone, and here we develop unbiased micro-computed tomography (micro-CT) methods to track the changes in tissue mineral content during cortical bone development; these methods are applicable to a wide range of applications in human and animal biology. We use them to identify not only morphological changes, but also, and for the first time, find an increase in bone material density with cortical maturation that occurs after the morphological character of the cortex has been formed. In addition, we show that IL-6 family cytokines have amplified and extended STAT3 phosphorylation responses in bone in the absence of SOCS3 and that deletion of gp130 in osteocytes rescues the features of delayed corticalisation in *Dmp1^Cre^:Socs3^f/f^* mice.

## Results

### Visualisation of cortical maturation between 12 and 15 weeks of age in murine femora and its delay in *Dmp1^{Cre}:Socs3^{f/f}* mice

There was no significant difference in femoral length between *Dmp1^{Cre}* and *Dmp1^{Cre}:Socs3^{f/f}* mice when scanned on a weekly basis (*Figure 1A,B*). A minor but statistically significant growth-related increase in femoral length was detected between 12 and 16 weeks in *Dmp1^{Cre}:Socs3^{f/f}* mice, consistent with a plateau of longitudinal growth by this time point (*Figure 1B*). From 12 to 16 weeks of

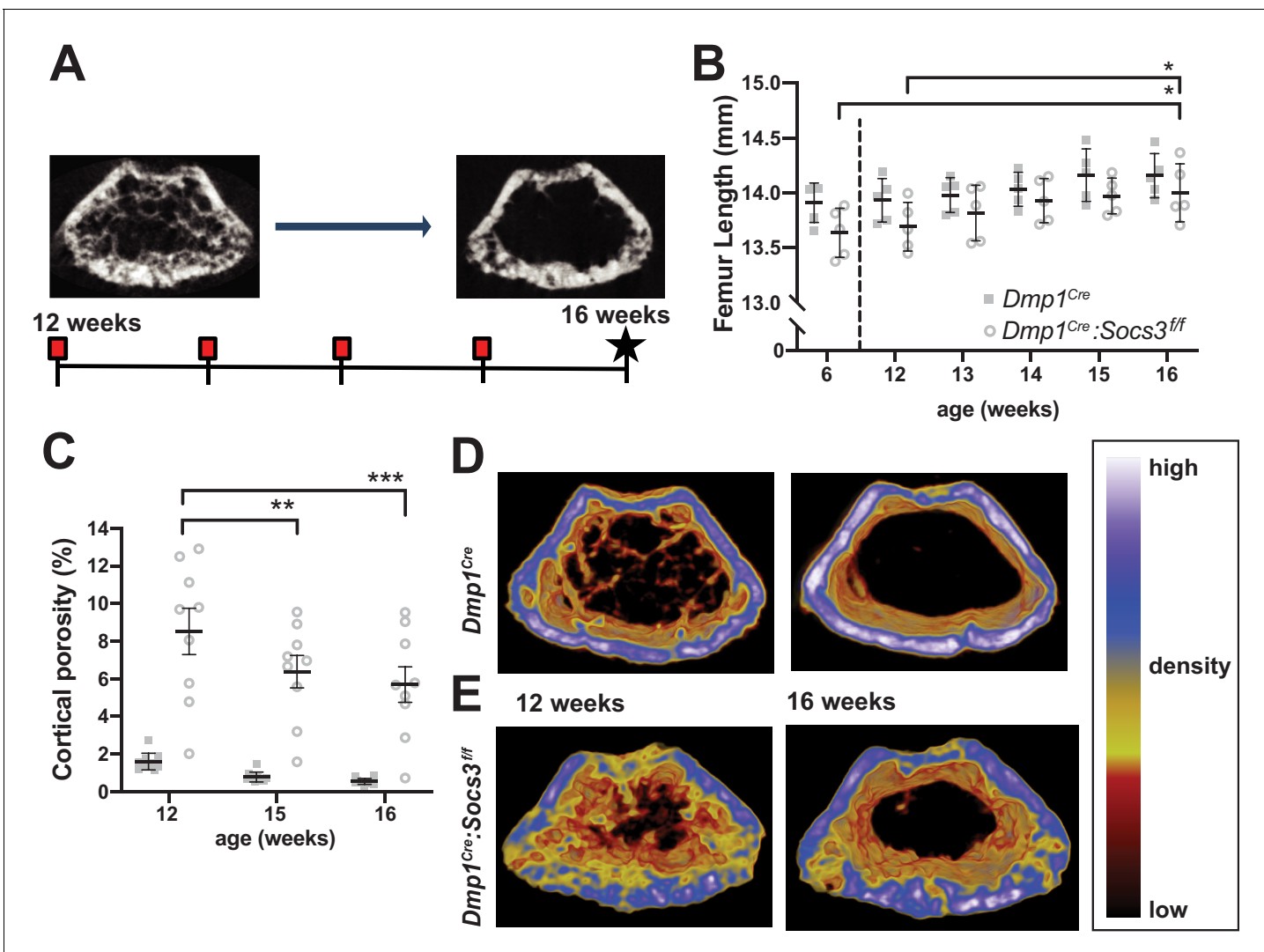

**Figure 1.** Transition to compact and highly mineralised bone is delayed in *Dmp1^{Cre}:Socs3^{f/f}* mice until longitudinal growth has ceased. (A) Schematic for studying time course of corticalisation in female *Dmp1^{Cre}:Socs3^{f/f}* mice. (B) Femur length at 6, and 12 to 16 weeks of age in female *Dmp1^{Cre}:Socs3^{f/f}* and control mice; *p<0.05 for comparison shown, by repeated measures two-way ANOVA and Tukey's post-hoc test; effect of age, p=0.0118; effect of genotype, p=0.1249 (not significant); age x genotype interaction, p=0.8443 (not significant); n = 5 mice/group. (C) Cortical porosity at 12, 15 and 16 weeks in female *Dmp1^{Cre}:Socs3^{f/f}* and control mice. **, p<0.01, ***p<0.001 for comparison shown (age-related change) by two-way ANOVA with repeated measures and Tukey post hoc test; effect of age, p<0.0001; effect of genotype, p<0.0001; age x genotype interaction, p=0.0429; no significant change in cortical porosity in control animals; n = 9 mice/group. (D) Representative micro-CT images with pseudo-colorization based on raw density of the micro-CT scans. These show the change in cortical morphology and increase in density within the metaphyseal region between 12 and 16 weeks: still images at the beginning and end of the sequence. Videos show transitions between all 5 images *Video 1*; *Video 2*.

The online version of this article includes the following source data and figure supplement(s) for figure 1:

**Source data 1.** Raw data for cortical porosity measurements.

**Figure supplement 1.** Cortical porosity of the femoral diaphysis of *Dmp1^{Cre}* and *Dmp1^{Cre}:Socs3^{f/f}* mice at 12 weeks of age.

age, as expected, the high cortical porosity was significantly reduced in *Dmp1^Cre^:Socs3^f/f^* mice, but did not reach the low normal levels seen in control (*Dmp1^Cre^*) mice (**Figure 1C**). Even though the more mature diaphysis has a lower cortical porosity than the metaphysis, cortical porosity was also elevated in the diaphysis of 12 week old *Dmp1^Cre^:Socs3^f/f^* mice (**Figure 1—figure supplement 1**).

Since longitudinal growth had reached a plateau, we merged bone density maps of the metaphyseal region to make videos showing changes in bone structure from 12 to 16 weeks of age **Video 1**; **Video 2** (still images at the beginning and end of the sequence are shown in **Figure 1D,E**). These indicated that, while control mice already have consolidated cortical bone in the metaphysis by 12 weeks of age, this structure was not fully formed even by 16 weeks of age in *Dmp1^Cre^:Socs3^f/f^* mice. In control mice, although the cortical shape was already formed, and cortical porosity did not reduce in this time period (**Figure 1C**), there was an increase in bone material density (shift in colour from blue to white, **Video 1**; **Figure 1D**). In *Dmp1^Cre^:Socs3^f/f^* mice, not only were cortical holes filled, consistent with quantitation in **Figure 1C**, but trabecular bone was remodelled into fewer elements, and some regions of bone in the cortex became more dense (shifting from yellow to blue to white in the pseudocolourised images) (**Video 2**; **Figure 1E**).

## Measuring the change in bone volumes of different densities by multi-level Otsu thresholding confirms the delay in *Dmp1^Cre^:Socs3^f/f^* mice

Bone analysis by micro-CT usually uses either manual or algorithm-based segmentation into its cortical and trabecular components on the basis of morphology (**Bouxsein et al., 2010**; **Buie et al., 2007**). However *Dmp1^Cre^:Socs3^f/f^* mice do not exhibit clearly distinguishable trabecular and cortical bone in the metaphysis at 12–15 weeks of age (**Cho et al., 2017**); instead they form a foamy network similar to that previously described during much earlier stages of cortical bone development (i.e. between birth and 14 days of age in C57BL/6 mice (**Bortel et al., 2015**). While cortical porosity reflected the changes that we were seeing, the standard manual or algorithm-based methods could not distinguish between cortical and trabecular bone in this model. We realised that the distinction of cortical and trabecular bone was possibly arbitrary in this situation. To overcome this, and to develop an objective way to quantify the changes that occur during cortical bone formation, we used unbiased multi-level Otsu thresholding (see Materials and Methods) of the entire metaphysis including all forms of bone to identify quantities of high-, mid-, and low-density bone, regardless of whether it could be defined as trabecular, cortical or undifferentiated. We used the more mature 15 week old control mice to define these three thresholds (**Figure 2A**). The thresholds were then applied to *Dmp1^Cre^* and *Dmp1^Cre^:Socs3^f/f^* femora at 12 and 15 weeks of age to quantify bone volume at each of those density levels across the metaphysis.

In *Dmp1^Cre^* mice, very little high density bone was detected in the metaphysis at 12 weeks of age (**Figure 2B**), consistent with the (non-calibrated) pseudocolourised images (**Figure 1D**). In the same animals, by 15 weeks of age, total bone volume (as a proportion of metaphyseal volume) was significantly increased. There were significant changes in all three densities of bone over the three week time period: the high density proportion increased approximately 14 fold, while the mid- and low-density bone were both significantly reduced by approximately 12% and 30%, respectively (**Figure 2B**). *Dmp1^Cre^:Socs3^f/f^* mice had more bone within the metaphysis (higher total bone volume/cross sectional area) than *Dmp1^Cre^* mice at both 12 and 15 weeks of age (**Figure 2B**). At 12 weeks, this was due to a two-fold greater proportion of low density bone. As *Dmp1^Cre^:Socs3^f/f^* mice aged, total bone volume did not change, but the proportion of low density bone decreased, while the proportion of high density bone increased (**Figure 2B**). This indicates that in both *Dmp1^Cre^* and *Dmp1^Cre^:Socs3^f/f^* mice, as the metaphysis ages, low density bone is replaced with higher density bone.

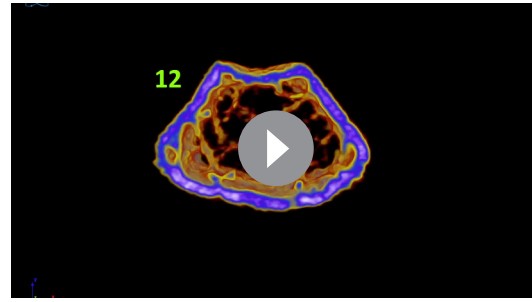

**Video 1.** Video showing the sequence of metaphyseal images obtained by micro-CT in control (*Dmp1^Cre^*) mice from 12 to 15 weeks of age. Pseudocolourisation reflects the same density scale shown in **Figure 1**.
https://elifesciences.org/articles/56666#video1

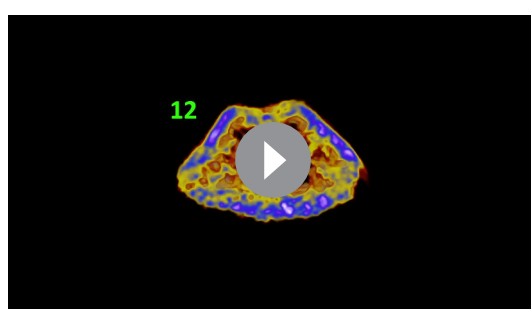

**Video 2.** Video showing the sequence of metaphyseal images obtained by micro-CT in *Dmp1^Cre^:Socs3^f/f^* mice from 12 to 15 weeks of age. Pseudocolourisation reflects the same density scale shown in *Figure 1*.
https://elifesciences.org/articles/56666#video2

## Measuring increasing proportions of high density bone at increasing distnace from the growth plate further confirms the delay in *Dmp1^Cre^:Socs3^f/f^* mice

Since new bone forms at the growth plate, we reasoned that the metaphyseal region as a whole would contain both immature and mature cortical bone, with the most mature bone at greatest distance from the growth plate (*Rauch, 2012*), and that this could be measured to identify differences in the corticalisation process between *Dmp1^Cre^* and *Dmp1^Cre^:Socs3^f/f^* mice. We t herefore assessed bone mass at the three levels of bone density working slice-by-slice from the proximal to the distal end of the femoral metaphysis (*Figure 3A*). At 12 weeks of age, *Dmp1^Cre^* mice exhibited a ~2.5 fold increase in mid-density bone (*Figure 3B*) at the proximal compared to the distal end (p<0.0001). Although there was very little high density bone present at 12 weeks of age, there was a significantly greater proportion in the last slice compared to the first 8 slices (~10 fold greater, p=0.037). Although the proportion of low density bone appeared to decline, this was not statistically significant. The proportion of bone within the cross sectional area increased with increasing distance from the growth plate (*Figure 3B*) due to narrowing of the metaphysis (*Figure 3—figure supplement 1*). Metaphyseal bone in the 12 week old murine femur therefore transitions from mainly low density, to mainly mid-density bone with increasing distance from the growth plate.

When *Dmp1^Cre^* control mice were assessed at 15 weeks of age along the metaphysis, the control bones exhibited an approximately 4-fold increase in the proportion of high density bone from the distal to the proximal end or the metaphysis, such that high density bone was the dominant type of bone present at the proximal end *Figure 3C*. This is clearly seen in the representative images, (compare right hand images in *Figure 3B and C*)). This shows that the transition from low to high density cortical bone that occurs as bone matures between 12 to 15 weeks can also be assessed at a single time point by a slice-by-slice analysis of the metaphysis, and that bone tissue matures even after cortical consolidation has occurred.

In *Dmp1^Cre^:Socs3^f/f^* mice at 12 weeks of age the shape of the density curve was distinctly different to that of age-matched *Dmp1^Cre^* mice (compare *Figure 3D* to *Figure 3B*). This was most notable in low density bone which was doubled in the proximal region compared to *Dmp1^Cre^* mice; this was statistically significant (*Figure 3—figure supplement 2* shows comparisons between genotypes at each slice and bone density level). The proportion of low density bone reduced significantly with distance from the growth plate, as mid-density bone increased (distance effect, p<0.0001 for both). There was no significant change in the very low proportion of high density bone, which was absent in most samples even in the proximal slices. This confirms the delayed cortical maturation in 12 week old *Dmp1^Cre^:Socs3^f/f^* mice. Their femora accumulate mid-density bone in regions furthest from the growth plate in the same manner as the control mice, but it is on the background of a high level of immature low density bone.

At 15 weeks of age the *Dmp1^Cre^:Socs3^f/f^* femora, like the *Dmp1^Cre^* femora, had more high density bone in regions furthest from the growth plate (*Figure 3E*). Although *Dmp1^Cre^:Socs3^f/f^* femora started with a lower level of high density bone than controls, they reached the same level of high density bone (*Figure 3—figure supplement 2*). *Dmp1^Cre^:Socs3^f/f^* femora at 15 weeks of age also had more low density bone than age-matched controls in the region closest to the growth plate, as at 12 weeks (*Figure 3—figure supplement 2*); and unlike age-matched *Dmp1^Cre^* controls, this declined significantly along the length of the bone. *Dmp1^Cre^:Socs3^f/f^* femora also had a higher proportion of mid-density bone compared to controls along almost the full length of the metaphysis (*Figure 3—figure supplement 2*). *Dmp1^Cre^:Socs3^f/f^* femora therefore exhibit both the reduction in cortical porosity and the increase in bone density associated with corticalisation. However, this

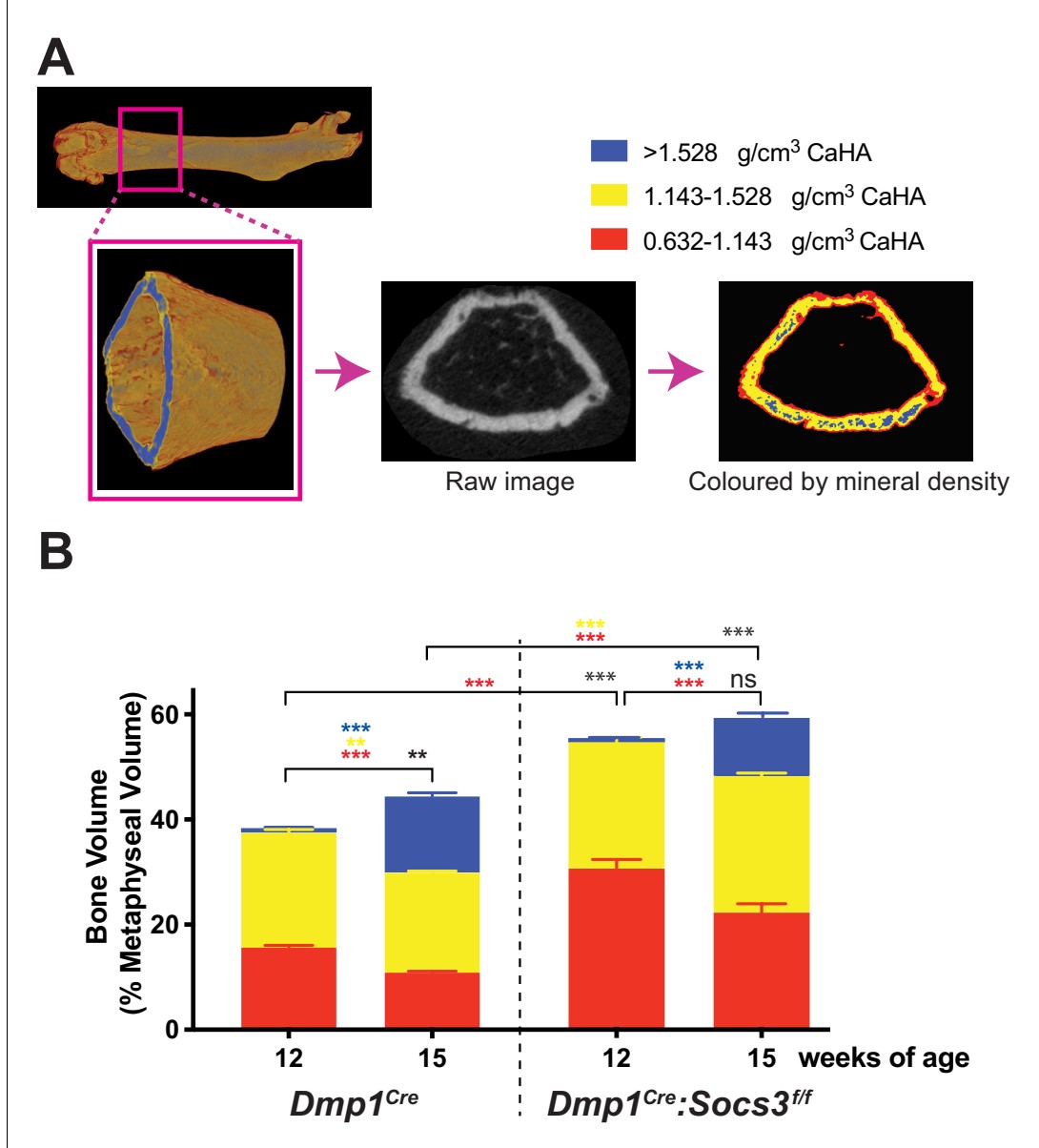

**Figure 2.** Transition to more highly mineralised bone during maturation of the metaphysis in female control (*Dmp1*<sup>Cre</sup>) and *Dmp1*<sup>Cre</sup>:*Socs3*<sup>f/f</sup> femora. (A) Diagram showing region measured, and Otsu thresholding of a raw micro-CT image. (B) Bone volume, as a percentage of total metaphyseal volume, segregated by low, mid and high density volumes, at 12 and 15 weeks in *Dmp1*<sup>Cre</sup> and *Dmp1*<sup>Cre</sup>:*Socs3*<sup>f/f</sup> mice; black asterisks denote significant changes in total bone volume, as indicated by square brackets. Changes in low-, mid- and high-density bone are indicated by coloured asterisks; error bars shown are SEM for the low-, mid- and high-density bone volumes. **, p<0.01; ***, p<0.001 for comparisons shown, determined by repeated measures two-way ANOVA with Šidák post-hoc test; n = 9–11 mice per group.

The online version of this article includes the following source data for figure 2:

**Source data 1.** Raw data for bone density measurements at all three densities.

process occurs in the presence of a greater quantity of low density bone than in control mice of the same age and sex. This confirms that *Dmp1*<sup>Cre</sup>:*Socs3*<sup>f/f</sup> mice have immature cortical bone compared with their age-matched controls.

The less mature nature of *Dmp1*<sup>Cre</sup>:*Socs3*<sup>f/f</sup> bone was also clear from pseudocolourised micro-CT images of the most mature regions of bone, which showed more mid-density bone than controls (compare images on the right of *Figure 1B and D*). When longitudinal femoral sections of these bones were taken at 16 weeks old and assessed by Ploton silver staining for osteocyte canaliculi and

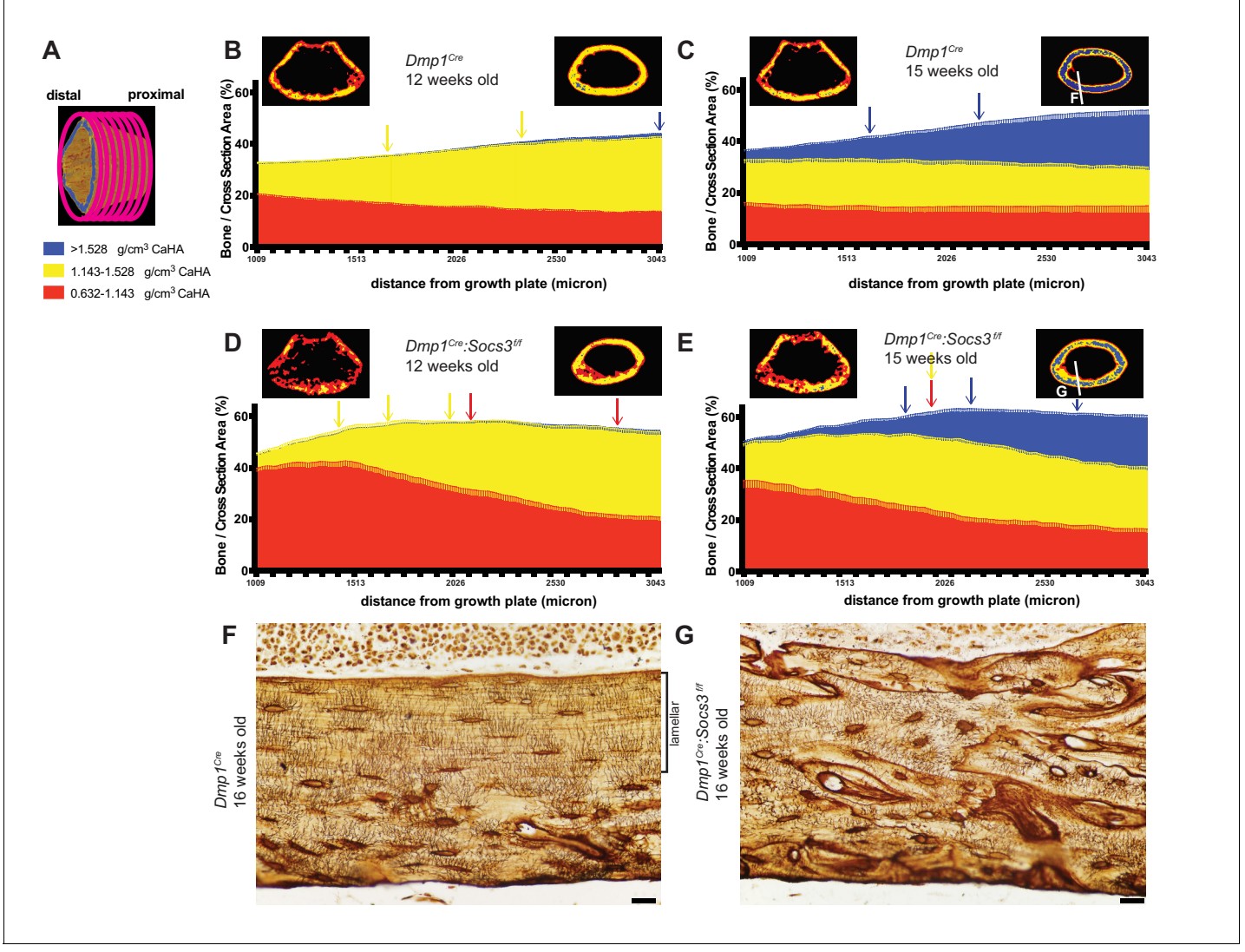

**Figure 3.** Bone becomes more mineralized with age, and with increasing distance from the growth plate in *Dmp1^Cre^* mice; this is delayed in *Dmp1^Cre^:Socs3^f/f^* mice. (A) The metaphyseal region was analysed in consecutive 9 micron slices. (B–E) Bone volume at low-, mid-, and high-density mineral levels in each slice from the distal to proximal end of the metaphyseal region at 12 weeks (B,D) and 15 weeks (C,E) of age in *Dmp1^Cre^* (B,C) and *Dmp1^Cre^:Socs3^f/f^* mice (D,E). Values are mean+ SEM, n = 9–11 mice per group; arrows denote where data becomes significantly different to the furthest slice from the arrow; colours denote the density level of bone that has changed. Pseudo-colourised images based on the Otsu thresholds show a representative sample for the top and bottom slice for each graph. Differences between genotypes are shown in *Figure 3—figure supplement 2*. (F,G) Ploton silver stain for osteocyte canaliculi and cement lines at the base of the metaphyseal region cut through the femur as indicated in the inset images of C and E for *Dmp1^Cre^* (F) and *Dmp1^Cre^:Socs3^f/f^* mice (G). Scale bar = 20 micron.

The online version of this article includes the following source data and figure supplement(s) for figure 3:

**Source data 1.** Bone volume at low-, mid-, and high-density mineral levels in each slice from the distal to proximal end of the metaphyseal region at 12 weeks and 15 weeks in *Dmp1^Cre^* and *Dmp1^Cre^:Socs3^f/f^* mice.
**Figure supplement 1.** Total bone area of the metaphyseal slices in *Dmp1^Cre^* and *Dmp1^Cre^:Socs3^f/f^* mice at 12 and 15 weeks of age (B,D,F).
**Figure supplement 2.** Comparison of high density, (A,B), mid-density (C,D) and low density (E,F) bone between *Dmp1^Cre^* and *Dmp1^Cre^:Socs3^f/f^* mice at 12 (A,C,E) and 15 weeks of age (B,D,F).
**Figure supplement 3.** Direct comparison of archived scans from *Dmp1^Cre^* and *Dmp1^Cre^:Socs3^f/f^* mice at 26 weeks of age.

cement lines (*Gaudin-Audrain et al., 2008*), it was clear that the nature of the bone material was different. At this site, in approximately 40% of the bone closest to the endocortical edge, control bone exhibited mature lamellar bone, characterised by flattened and aligned osteocytes with dendritic processes primarily running perpendicular to the endocortical surface (*Figure 3F*). In contrast,

osteocytes in *Dmp1^Cre^:Socs3^f/f^* bone at the same anatomical location were more rounded, showed no ordered orientation, and the bone contained many cement lines (*Figure 3G*). Since we have previously reported that the newly forming *Dmp1^Cre^:Socs3^f/f^* cortical bone does not contain cartilage remnants (*Cho et al., 2017*), the abundant cement lines of *Dmp1^Cre^:Socs3^f/f^* cortical bone do not reflect retention of immature endochondral cartilage and woven bone. Rather, it reflects rapid remodelling and replacement of that structure with woven bone. Although the control cortical bone had accrued lamellar bone at this time point, this has not occurred in the *Dmp1^Cre^:Socs3^f/f^* cortex. This confirms our earlier assertion that cortical bone maturation is delayed in these mice (*Cho et al., 2017*).

We also analysed archived micro-CT femoral scans from male and female 26 week old *Dmp1^Cre^: Socs3^f/f^* mice which we previously reported to have normalised cortical morphology, but impaired strength in females only (*Cho et al., 2017*). Using our three-tiered density analysis, we observed that although female *Dmp1^Cre^:Socs3^f/f^* femora had normal bone mass, they still had significantly less high- and mid- density bone, and more low-density bone than controls (*Figure 3—figure supplement 3*).

## STAT3 phosphorylation in response to IL-6 family cytokines is prolonged in *Dmp1^Cre^:Socs3^f/f^* bone

Having established two methods to quantify cortical maturation, we then sought to identify the hyperactive cytokine responsible for the *Dmp1^Cre^:Socs3^f/f^* phenotype by knocking down candidate receptors. SOCS3 provides negative feedback for leptin receptor, G-CSF receptor, and gp130 (*Morris et al., 2018*), which have all been reported to modify bone formation. IL-6 was not tested, since we have previously reported that the *Dmp1^Cre^:Socs3^f/f^* phenotype is not rescued by IL-6 deletion (*Cho et al., 2017*). When tested in a calvarial injection model, *Dmp1^Cre^:Socs3^f/f^* mice showed prolonged STAT3 phosphorylation in response to all IL-6 family cytokines tested (IL-11, LIF and OSM) compared to *Dmp1^Cre^* mice (*Figure 4*). In contrast, there was no STAT3 phosphorylation response to either Leptin or G-CSF (data not shown). The *Dmp1^Cre^:Socs3^f/f^* phenotype may therefore result from prolonged STAT3 phosphorylation downstream of gp130 binding cytokines, including IL-11, LIF and OSM.

There was no difference in basal phospho-STAT3 levels detected by western blot, consistent with observations in other cell-specific knockouts of SOCS3 (*Croker et al., 2003*; *Croker et al., 2004*). This is likely due to the low level of STAT3 phosphorylation in basal conditions relative to the stimulated response, or may reflect that the tissue contains a mixture of cell types, including cells with normal levels of SOCS3. STAT3 phosphorylation was elevated in osteocytes by immunohistochemical quantitation of phospho-STAT3 positive osteocytes, which were significantly elevated in both male and female tibial samples (*Figure 4D*), confirming that a lack of SOCS3 negative feedback results in elevated STAT3 signalling in response to endogenous stimuli.

## Development of a new mouse with *Dmp1^Cre^* -targeted deletion of gp130

We previously reported that the *Dmp1^Cre^:Socs3^f/f^* phenotype is not rescued by IL-6 deletion (*Cho et al., 2017*), and since all other gp130-binding cytokines tested induced prolonged phosphorylation, we next determined whether gp130 deletion in osteocytes might resolve the phenotype. For this, we developed a new *Il6st-flox* mouse. The rationale for this was that the gp130-flox mouse we previously utilised (*Standal et al., 2014*; *Johnson et al., 2014*) is reported to increase release of soluble gp130 by targeted cells (*Betz et al., 1998*), which might act locally as an inhibitor of signalling in other cell types.

The new mouse with *Dmp1^Cre^* targeted deletion (*Dmp1^Cre^:Il6st^f/f^*) showed reduced *Il6st* transcription in bone (*Table 1*) and exhibited a similar phenotype to that observed with the previously published gp130 targeting strategy (*Dmp1^Cre^:gp130^f/f^*). That is, the mouse exhibited low trabecular bone mass, increased periosteal circumference, and reduced bone formation with no reduction in bone resorption (*Table 1*; *Standal et al., 2014*; *Johnson et al., 2014*). This confirmed our earlier finding that gp130 in osteocytes is required for normal bone formation and normal trabecular bone mass.

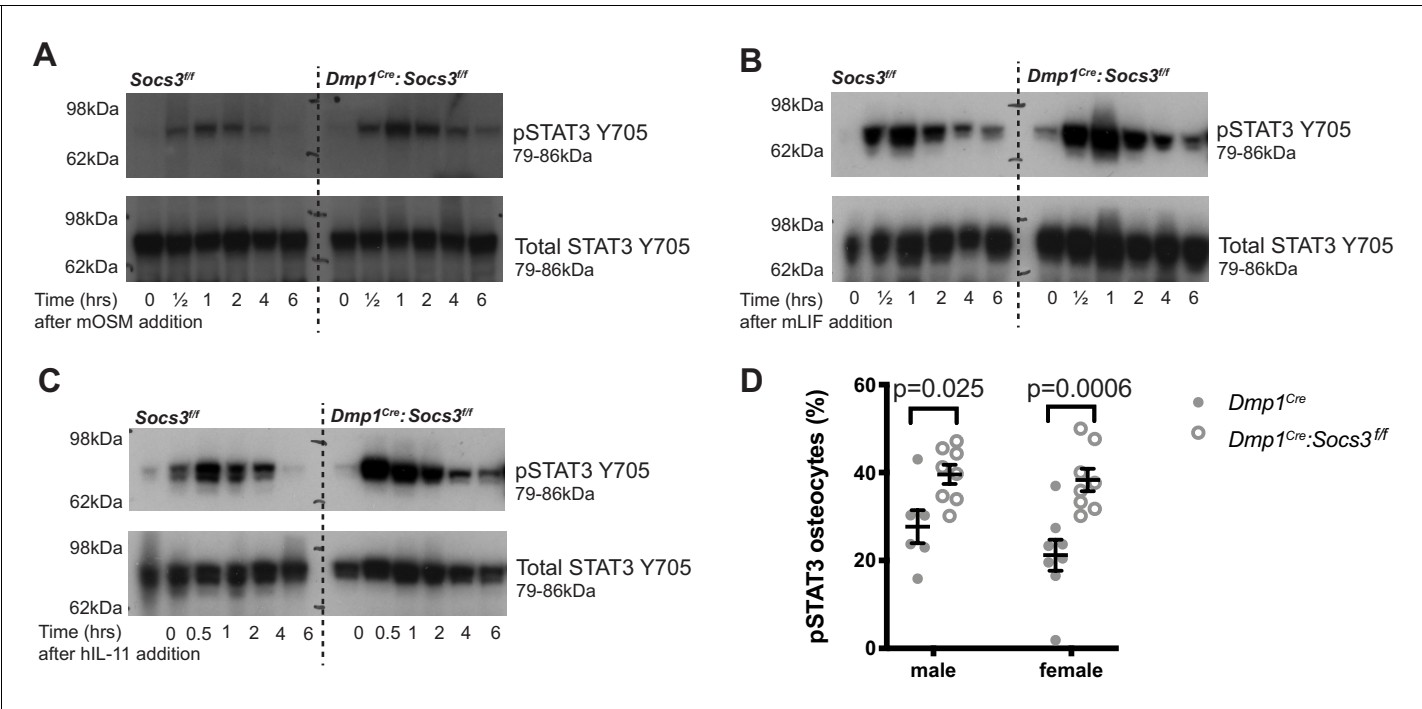

**Figure 4.** STAT3 phosphorylation response is magnified and extended in calvarial bone from male *Socs3^f/f* (control) and *Dmp1^Cre:Socs3^f/f* mice treated with gp130-dependent stimuli (A-C), and elevated under basal conditions in *Dmp1^Cre:Socs3^f/f* osteocytes (D). (A-C) Mice were given a single calvarial injection of murine oncostatin M (mOSM) (A), murine leukaemia inhibitory factor (mLIF) (B), or human interleukin- 11 (hIL-11) (C). Calvariae were collected at the time of injection, and at 30 min, 1, 2, 4, and 6 hr. Shown are phospho- and total STAT3 westeArn blots; 4 mice were assessed per group; representative blots shown. (D) Quantitation of phospho-STAT3 (pSTAT3) immunohistochemistry in the tibial lateral metaphyses of 12 week old male and female *Dmp1^Cre* and *Dmp1^Cre:Socs3^f/f* mice; n = 5–8 per group; values are individual points with mean and SEM; p values generated by two-way ANOVA with Sidak post-hoc test.

The online version of this article includes the following source data for figure 4:

**Source data 1.** Uncropped gels for phospho-STAT3 (left panels) and total STAT3/pan actin (right panels) for calvarial samples from *Socs^f/f* and *Dmp1^Cre:Socs3^f/f* mice treated with OSM (A), LIF (B) or IL-11 (C).

## Rescue of the *Dmp1^Cre:Socs3^f/f* phenotype by *Dmp1^Cre* -targeted deletion of gp130

When *Dmp1^Cre:Il6st^f/f* mice, which themselves had normal cortical porosity (*Figure 5A*) were crossed with *Dmp1^Cre:Socs3^f/f* mice, the resulting *Dmp1^Cre:Socs3^f/f:Il6st^f/f* mice exhibited normalised cortical porosity at 12 weeks of age in both male and female mice (*Figure 5B,C*). Normalisation was also observed in bone volume segregated by density. Levels of low- and mid-density bone were each significantly reduced in *Dmp1^Cre:Socs3^f/f:Il6st^f/f* mice compared with the high levels observed in *Dmp1^Cre:Socs3^f/f* mice; this was observed in both males and females (*Figure 5D*).

When the density profiles were assessed along the length of the bone, female *Dmp1^Cre:Socs3^f/f: Il6st^f/f* mice showed a pattern that was very similar to wild type mice of the same age (compare *Figure 5E* to *Figure 3A*). The high volume of low-density bone seen in *Dmp1^Cre:Socs3^f/f* mice at the distal end (*Figure 5E*) was not observed in *Dmp1^Cre:Socs3^f/f:Il6st^f/f* mice (*Figure 5F*). When compared directly (*Figure 5—figure supplement 1A*) the proportion of low density bone was significantly lower in *Dmp1^Cre:Socs3^f/f:Il6st^f/f* mice compared to *Dmp1^Cre:Socs3^f/f* mice in this region. The level of mid-density bone was also significantly lower in *Dmp1^Cre:Socs3^f/f:Il6st^f/f* mice compared to *Dmp1^Cre:Socs3^f/f* mice (*Figure 5—figure supplement 1B*), indicating a return to a phenotype similar to control mice (compare with *Figure 3—figure supplement 2D*). The proportion of high density bone increased in both *Dmp1^Cre:Socs3^f/f* and *Dmp1^Cre:Socs3^f/f:Il6st^f/f* female femora (*Figure 5E,F*). These results indicate that deletion of gp130 from osteocytes normalises the delayed corticalisation of female *Dmp1^Cre:Socs3^f/f* mice.

**Table 1.** Phenotypic data from 12 week old $Dmp1^{Cre}$:$Il6st^{f/f}$ mice, including mRNA levels of the targeted gene in flushed femoral samples ($Il6st$:$B2m$), femoral trabecular and cortical structure (by micro-computed tomography) and tibial trabecular histomorphometry in the secondary spongiosa.

Data are mean ± SEM. *, p<0.05, **, p<0.01; ***, p<0.001 vs sex- and age-matched controls ($Dmp1Cre$) by two-way ANOVA with Šidák post-hoc test.

| Parameter | male | | female | |
|---|---|---|---|---|
| | $Dmp1^{Cre}$ | $Dmp1^{Cre}$:$Il6st^{f/f}$ | $Dmp1^{Cre}$ | $Dmp1^{Cre}$:$Il6st^{f/f}$ |
| Number of samples | 7 | 10 | 6 | 6 |
| $IL6st$:$B2m$ (mRNA) | 1.14 ± 0.21 | 0.52 ± 0.08** | 0.88 ± 0.15 | 0.40 ± 0.18 |
| Trabecular bone volume (%) | 13.72 ± 1.26 | 8.24 ± 0.96** | 10.82 ± 1.06 | 6.06 ± 0.51** |
| Trabecular number (/mm) | 2.54 ± 0.21 | 1.27 ± 0.12*** | 2.38 ± 0.22 | 1.12 ± 0.08*** |
| Trabecular Thickness (µm) | 53.5 ± 0.8 | 64.1 ± 2.6*** | 45.6 ± 1.0 | 54.0 ± 1.3*** |
| Trabecular Separation (µm) | 217 ± 14.01 | 469 ± 26*** | 239 ± 15 | 415 ± 29*** |
| Periosteal Perimeter (mm) | 7.21 ± 0.14 | 8.14 ± 0.25*** | 6.78 ± 0.13 | 7.29 ± 0.10* |
| Marrow Area (mm$^2$) | 1.10 ± 0.04 | 1.37 ± 0.08*** | 0.94 ± 0.04 | 1.11 ± 0.02* |
| Moment of Inertia (mm$^4$) | 0.395 ± 0.021 | 0.566 ± 0.51*** | 0.344 ± 0.02 | 0.404 ± 0.009 |
| Bone Formation Rate/Bone Surface (%) | 0.454 ± 0.019 | 0.295 ± 0.048** | 0.704 ± 0.03 | 0.481 ± 0.065* |
| Double labelled Surface/Bone Surface (%) | 29.1 ± 1.8 | 20.1 ± 2.8** | 26.6 ± 1.5 | 15.1 ± 1.8** |
| Mineral Appositional Rate (µm/day) | 1.30 ± 0.08 | 1.10 ± 0.14 | 1.98 ± 0.24 | 1.63 ± 0.19 |
| Osteoclast Surface/Bone Surface (%) | 4.39 ± 0.58 | 6.59 ± 0.89 | 3.61 ± 0.45 | 4.82 ± 0.83 |

A similar, though less pronounced normalisation effect was observed in male $Dmp1^{Cre}$:$Socs3^{f/f}$:$Il6st^{f/f}$ mice compared to $Dmp1^{Cre}$:$Socs3^{f/f}$ mice (*Figure 5G,H*). Male $Dmp1^{Cre}$:$Socs3^{f/f}$ mice exhibited a lesser delay in corticalisation compared to females in our earlier study (*Cho et al., 2017*). $Dmp1^{Cre}$:$Socs3^{f/f}$ mice, like females of the same genotype, exhibited a reduction in low density bone, and an increase in both mid-density and high-density bone along the length of the metaphysis (*Figure 5G*). When gp130 was deleted from osteocytes, the male $Dmp1^{Cre}$:$Socs3^{f/f}$:$Il6st^{f/f}$ mice exhibited the same differences compared to sex-matched $Dmp1^{Cre}$:$Socs3^{f/f}$ mice as seen in the females: a lower level of low density and mid-density bone, and no difference in high density bone (*Figure 5—figure supplement 1D–F*). This indicates that cortical consolidation specifically requires SOCS3 mediated suppression of IL-6 family cytokine signalling within osteocytes in both male and female mice.

To determine the mechanism by which the cortical bone consolidation was improved through deletion of gp130 in osteocytes, we carried out histology. There was no clear reduction in calcein labels in the metaphyseal cortex in $Dmp1^{Cre}$: $Socs3^{f/f}$:$Il6st^{f/f}$ mice compared to $Dmp1^{Cre}$:$Socs3^{f/f}$ mice (*Figure 6A,B*). Extensive double-labelled surfaces were observed in both genotypes in this region, but more cortical pores (containing blood vessels and sometimes bone marrow) were seen in the $Dmp1^{Cre}$:$Socs3^{f/f}$ mice. When quantified, the pore area was reduced in $Dmp1^{Cre}$:$Socs3^{f/f}$:$Il6st^{f/f}$ bone by approximately 60% (*Figure 6C*). Both osteoclasts and osteoblasts were detected within the pores in $Dmp1^{Cre}$:$Socs3^{f/f}$ bone, and due to the reduction in pore area, the number of osteoclasts and osteoblasts were lower in $Dmp1^{Cre}$:$Socs3^{f/f}$:$Il6st^{f/f}$ bone than $Dmp1^{Cre}$:$Socs3^{f/f}$ bone (*Figure 6D, E*). When we assessed osteoclasts by TRAP staining, voids in $Dmp1^{Cre}$:$Socs3^{f/f}$ cortical bone exhibited large highly active osteoclasts within the newly forming cortex (*Figure 6F,G*). Although murine osteoclasts do not usually demonstrate clear resorption pits, they were observed in this region in the $Dmp1^{Cre}$:$Socs3^{f/f}$ metaphyses (*Figure 6J*).

In contrast to the increased resorption observed in the tibial cortex in $Dmp1^{Cre}$:$Socs3^{f/f}$ metaphyses, the same region in $Dmp1^{Cre}$:$Socs3^{f/f}$:$Il6st^{f/f}$ mice rarely exhibited osteoclasts within the cortical pores (*Figure 6H,I*). When quantified, the voids within the metaphyseal cortical bone of $Dmp1^{Cre}$: $Socs3^{f/f}$ mice had greater osteoclast numbers than in $Dmp1^{Cre}$:$Socs3^{f/f}$:$Il6st^{f/f}$ mice, yet there was no difference in osteoblast numbers between the genotypes (*Figure 6K,L*). Analysis of flushed femoral samples, including mainly cortical bone provided some additional evidence of suppressed osteoclast formation in $Dmp1^{Cre}$:$Socs3^{f/f}$:$Il6st^{f/f}$ mice (*Table 2*); $Acp5$ (the gene for TRAP) was significantly less in these samples compared to $Dmp1^{Cre}$:$Socs3^{f/f}$ mice. This was not associated with a significant change in RANKL mRNA ($Tnfsf11$), but OPG mRNA ($Tnfrsf11b$) levels were mildly, and barely statistically significantly, greater in samples from $Dmp1^{Cre}$:$Socs3^{f/f}$:$Il6st^{f/f}$ mice.

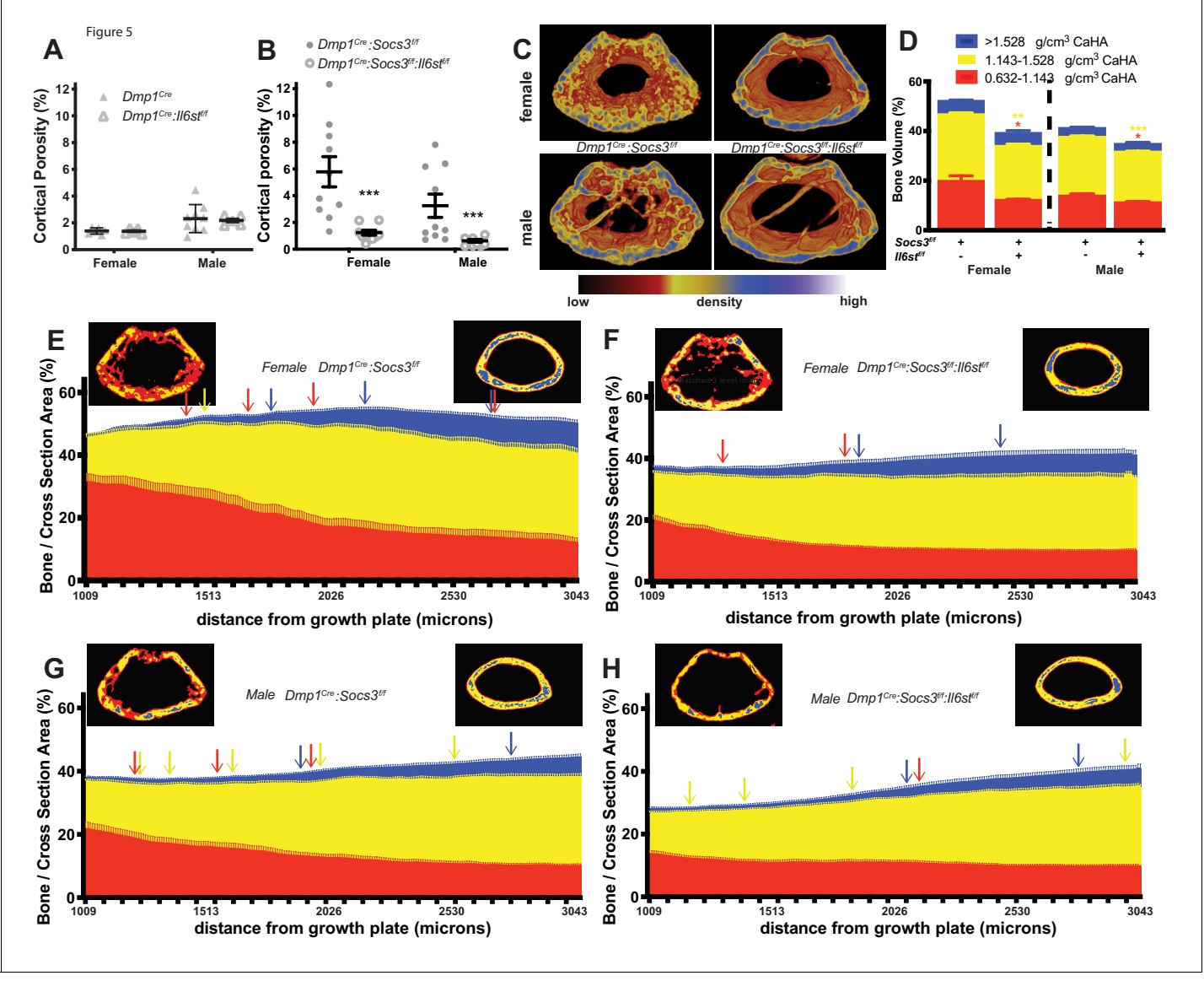

**Figure 5.** Rescue of delayed corticalisation in 12 week old male and female *Dmp1^Cre^:Socs3^f/f^:Il6st^f/f^* mice. Metaphyseal cortical porosity in female and male control (Dmp1Cre) and *Dmp1^Cre^:Il6st^f/f^* mice (**A**) and *Dmp1^Cre^:Socs3^f/f^* and *Dmp1^Cre^:Socs3^f/f^:Il6st^f/f^* mice (**B**) and representative Otsu-based pseudocolorised images (**C**). (**D**) Bone volume, as a percentage of total metaphyseal volume, segregated by low, mid and high density volumes, in 12 week old *Dmp1^Cre^:Socs3^f/f^* and *Dmp1^Cre^:Socs3^f/f^:Il6st^f/f^* mice. Changes in low-, mid- and high-density bone are indicated by coloured asterisks; error bars shown are SEM for the low-, mid- and high-density bone volumes. *, p<0.5; **, p<0.01; ***, p<0.001 for comparisons shown, determined by repeated measures two-way ANOVA with Šidák post-hoc test; n = 9–11 mice per group. (**E–H**) Bone volume at low-, mid-, and high-density mineral levels in each slice from the distal to proximal end of the metaphyseal region in 12 week old female (**E,F**) and male (**G,H**) *Dmp1^Cre^:Socs3^f/f^* femora (**E,G**) and *Dmp1^Cre^:Socs3^f/f^:Il6st^f/f^* (**F,H**) f. Values are mean+ SEM, n = 9–11 mice per group; arrows denote where data becomes significantly different to the furthest slice from the arrow; colours denote the density level of bone that has changed. Pseudo-colourised images show a representative sample for the top and bottom slice for each graph. Differences between genotypes are shown in *Figure 5—figure supplement 1* .
The online version of this article includes the following source data and figure supplement(s) for figure 5:

**Source data 1.** Bone volume at low-, mid-, and high-density mineral levels in each slices from the distal to proximal end of the metaphyseal region in 12 week old female and male *Dmp1^Cre^:Socs3^f/f^* mice and *Dmp1^Cre^:Socs3^f/f^:Il6st^f/f^* mice.

**Figure supplement 1.** Direct comparison of high density (**A,B**), mid-density (**C,D**) and low-density (**E,F**) bone between *Dmp1^Cre^:Socs3^f/f^* and *Dmp1^Cre^:Socs3^f/f^:Il6st^f/f^* mice.

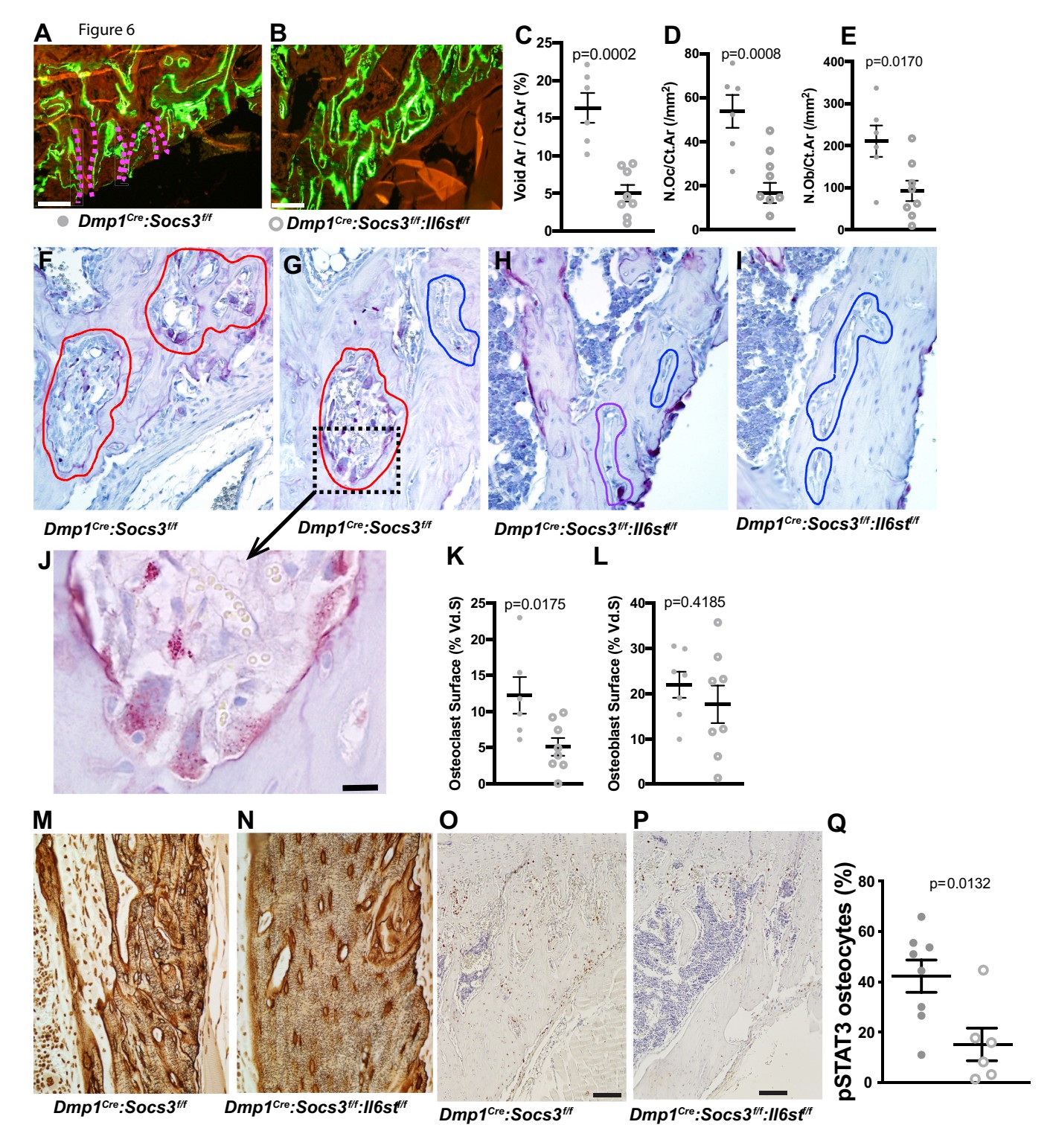

**Figure 6.** Closure of cortical pores in *Dmp1^{Cre}:Socs3^{f/f}* mice by *Il6st* knockdown involves reduced cortical resorption and reduced STAT3 phosphorylation. (A,B) Representative calcein labelling in *Dmp1^{Cre}:Socs3^{f/f}* and *Dmp1^{Cre}:Socs3^{f/f}:Il6st^{f/f}* mice. Pink dashed lines show transcortical pores. (C–E) Histomorphometry of cortical void area (Vd.Ar/Ct.Ar), numbers of osteoclasts (N.Oc/Ct.Ar) and osteoblasts (N.Ob/Ct.Ar) each normalised for cortical area. (F–I) Sections of the upper (F,H) and lower (G,I) regions of the tibial lateral proximal metaphysis showing more osteoclastic resorption (outlined in red) in *Dmp1^{Cre}:Socs3^{f/f}* compared to *Dmp1^{Cre}:Socs3^{f/f}:Il6st^{f/f}* cortex; a high resolution image of resorption pits from boxed region of panel F is shown in panel J. (K,L) Cortical osteoclast surface and osteoblast surface normalised to the extent of void surface in 12 week old *Dmp1^{Cre}:Socs3^{f/f}*

*Figure 6 continued on next page*

*Figure 6 continued*

and *Dmp1^Cre^:Socs3^f/f^:Il6st^f/f^* mice. (M,N) Ploton silver stain showing loss of the extensive cement lines and more ordered orientation of osteocyte cell bodies in the 12 week old tibial lateral metaphysis in *Dmp1^Cre^:Socs3^f/f^:Il6st^f/f^* mice compared to *Dmp1^Cre^:Socs3^f/f^* mice. (O–Q) Phospho-STAT3 (pSTAT3) immunohistochemistry in the tibial lateral metaphysis of 12 week old *Dmp1^Cre^:Socs3^f/f^* (O) and *Dmp1^Cre^:Socs3^f/f^:Il6st^f/f^* mice (P) and osteocyte quantitation (Q) in the same region. For all graphs, data are individual data points with mean ± SEM. P values determined by Student's t-test.

The online version of this article includes the following figure supplement(s) for figure 6:

**Figure supplement 1.** Additional histological images, including analysis of calvarial bone.

To determine whether the cortical bone-specific effect on osteoclast formation could be detected at a second site, we carried out a preliminary analysis of calvarial bone from 12 week old *Dmp1^Cre^:Socs3^f/f^* and *Dmp1^Cre^* mice collected in our original study (*Figure 6—figure supplement 1B*). There we observed a greater proportion of marrow-filled area within the calvarial bone (diploe) in *Dmp1^Cre^:Socs3^f/f^* mice, and both a high level of calcein labelling, and abundant osteoclasts (*Figure 6—figure supplement 1C,D*). This confirms that the stimulatory effect of SOCS3 deletion on bone resorption occurs also in this cortical site that has not formed through trabecular consolidation.

The higher level of resorption in *Dmp1^Cre^:Socs3^f/f^* cortical bone was reflected in Ploton silver stains carried out at a slightly more distal location in the tibia. Indeed, in this more mature region, *Dmp1^Cre^:Socs3^f/f^:Il6st^f/f^* mice exhibited a more mature osteocyte canalicular network than *Dmp1^Cre^:Socs3^f/f^* (*Figure 6M,N*). This was indicated by a greater proportion of osteocyte cell bodies aligned parallel to the bone surface, and less cement lines within the cortex. The normalized structure resembled the canalicular network of *Dmp1^Cre^* mice in the same region (*Figure 6—figure supplement 1*). This normalization of cortical bone structure and maturation was associated with a significant reduction in phospho-STAT3 staining in osteocytes within the newly forming cortical bone (*Figure 6O–Q*). This indicates that for cortical consolidation to occur in the metaphysis, bone resorption must be suppressed, and this can be achieved by reducing gp130-dependent STAT3 phosphorylation in osteocytes.

We previously reported lower mechanical strength of the femoral diaphysis of 26 week old *Dmp1^Cre^:Socs3^f/f^* mice compared to *Dmp1^Cre^* mice (tested by three point bending) (*Cho et al., 2017*). This included a 50% reduction in post-yield deformation (*Cho et al., 2017*). In the present study, when *Dmp1^Cre^:Socs3^f/f^* mice were crossed with *Il6st^f/f^* mice, this parameter was approximately doubled in both male and female mice (*Table 3*), representing a significant improvement. Male mice *Dmp1^Cre^:Socs3^f/f^:Il6st^f/f^* mice also exhibited a significant reduction in yield force. This suggests that the bone of *Dmp1^Cre^:Socs3^f/f^:Il6st^f/f^* mice is more ductile. While this prevents the bones from fracturing earlier (reflected in the low yield force in the male mice), this does not prevent the bone from fracturing at the same force as the *Dmp1^Cre^:Socs3^f/f^* bone. Notably, in our earlier study, we also did not see a change in the ultimate force, indicating that the defect in corticalisation changes bone

**Table 2.** mRNA analysis of osteoclast-related genes in flushed femora from 12 week old *Dmp1^Cre^:Socs3^f/f^* and *Dmp1^Cre^:Socs3^f/f^:Il6st^f/f^*.

Data are mean ± SEM normalised to the geometric mean of B2m and Hprt1; p values from two-way ANOVA analysis are shown in the right column, including p values for the comparison of male vs female, *Dmp1^Cre^:Socs3^f/f^* and *Dmp1^Cre^:Socs3^f/f^:Il6st^f/f^* (genotype) and the interaction between the two. No statistical differences between genotypes were detected by Šidák post-hoc test within each sex.

| Parameter | male | | female | | p values |
|---|---|---|---|---|---|
| | *Dmp1^Cre^:Socs3^f/f^* | *Dmp1^Cre^:Socs3^f/f^:Il6st^f/f^* | *Dmp1^Cre^:Socs3^f/f^* | *Dmp1^Cre^:Socs3^f/f^:Il6st^f/f^* | |
| Number of samples | 10 | 8 | 10 | 8 | |
| *Tnfsf11* (RANKL) | 0.015 ± 0.002 | 0.016 ± 0.004 | 0.015 ± 0.003 | 0.015 ± 0.003 | Male vs female: 0.752; Genotype: 0.841; Interaction: 0.923. |
| *Tnfrsf11b* (OPG) | 0.067 ± 0.014 | 0.097 ± 0.017 | 0.026 ± 0.004 | 0.044 ± 0.009 | **Male vs female: 0.0003; Genotype: 0.048;** Interaction: 0.635. |
| *Acp5* (TRAP) | 3.64 ± 0.36 | 1.94 ± 0.26 | 10.25 ± 1.45 | 7.41 ± 1.12 | **Male vs female:<0.0001; Genotype: 0.028;** Interaction: 0.568. |

**Table 3.** Results of three point bending tests of femora from male and female 12 week old $Dmp1^{Cre}$:$Socs3^{f/f}$ and $Dmp1^{Cre}$:$Socs3^{f/f}$:$Il6st^{f/f}$ mice.

Data is mean ± SEM. *, p<0.05; **, p<0.01 vs $Dmp1^{Cre}$:$Socs3^{f/f}$ by two-way ANOVA with Šidák post-hoc test.

| Parameter | male | | female | |
|---|---|---|---|---|
| | $Dmp1^{Cre}$:$Socs3^{f/f}$ | $Dmp1^{Cre}$:$Socs3^{f/f}$:$Il6st^{f/f}$ | $Dmp1^{Cre}$:$Socs3^{f/f}$ | $Dmp1^{Cre}$:$Socs3^{f/f}$:$Il6st^{f/f}$ |
| Number of samples | 9 | 11 | 10 | 9 |
| Ultimate Force (N) | 16.48 ± 0.69 | 15.31 ± 0.60 | 13.40 ± 0.25 | 13.79 ± 0.59 |
| Ultimate Deformation (mm) | 0.304 ± 0.025 | 0.330 ± 0.02 | 0.350 ± 0.029 | 0.350 ± 0.019 |
| Yield Force (N) | 15.44 ± 1.75 | **11.29 ± 0.70**\*\* | 11.03 ± 0.58 | 10.19 ± 0.69 |
| Yield Deformation (mm) | 0.240 ± 0.022 | 0.212 ± 0.022 | 0.282 ± 0.024 | 0.219 ± 0.026 |
| Post-Yield Deformation (mm) | 0.063 ± 0.018 | **0.126 ± 0.019**\* | 0.068 ± 0.013 | **0.152 ± 0.020**\*\* |
| Energy to Failure (mJ) | 2.61 ± 0.31 | 2.94 ± 0.25 | 2.34 ± 0.16 | **2.99 ± 0.19**\* |
| Ultimate Stress (MPa) | 47.30 ± 4.69 | 39.67 ± 1.72 | 63.96 ± 2.39 | 54.29 ± 2.33 |
| Ultimate Strain (%) | 0.033 ± 0.003 | 0.036 ± 0.003 | 0.034 ± 0.003 | 0.037 ± 0.002 |
| Yield Stress (MPa) | 41.65 ± 5.20 | **28.97 ± 1.46**\* | 52.46 ± 3.02 | 40.42 ± 3.44 |
| Yield Strain (%) | 0.026 ± 0.003 | 0.023 ± 0.003 | 0.028 ± 0.002 | 0.023 ± 0.003 |
| Post-Yield Strain (%) | 0.007 ± 0.002 | **0.014 ± 0.002**\* | 0.007 ± 0.001 | **0.016 ± 0.002**\*\* |

ductility, and behaviour under strain, but does not change its ultimate maximal strength. In addition, post-yield strain (which corrects the mechanical testing data for the size and shape of each individual bone sample) was significantly greater in both male and female $Dmp1^{Cre}$:$Socs3^{f/f}$:$Il6st^{f/f}$ mice compared to the mutant $Dmp1^{Cre}$:$Socs3^{f/f}$ mice (**Table 3**). This suggests that the rescue of the strength defect in these mice resulted from an improvement in the material quality of the cortical bone at the diaphysis, consistent with an improvement in cortical bone maturation.

## Discussion

This study shows that cortical bone formation includes both consolidation of structural elements and a transition from less organised and less mineralized bone to a mature lamellar structure; new methods were developed to show this. In addition, we show that consolidation, maturation, and mechanical performance of the cortical structure requires suppression of STAT3 phosphorylation downstream of gp130 signalling in osteocytes. When STAT3 phosphorylation is elevated, osteoclasts actively resorb the cortical structure, and formation of a lamellar structure is delayed. This process, outlined in **Figure 7**, provides new insight into the changes in bone that determine cortical bone structure and into the cytokine signalling pathways and cell types involved.

In earlier work, we reported delayed closure of cortical pores until after 12 weeks of age in female $Dmp1^{Cre}$:$Socs3^{f/f}$ mice (**Cho et al., 2017**). That was associated with an extremely high trabecular bone volume (which would have included unconsolidated cortical bone). This is consistent with the high total bone volume we have observed in the present study. Normally the transformation from a foamy woven bone cortex to a continuous cortical mass occurs between 10 and 14 days of age in C57BL/6 mice (**Bortel et al., 2015**). This coincides with rapid longitudinal growth, and occurs prior to development of the femoral elliptical shape. Since cortical consolidation in $Dmp1^{Cre}$:$Socs3^{f/f}$ mice occurred after both bone length and femoral shape are established (**Cho et al., 2017**), we reasoned that their metaphyses could be used to track cortical consolidation without the complication of coincident changes in bone size and shape.

By studying these mice, we observed that cortical consolidation includes both coalescence of trabecular bone to form a thickened cortical shell followed by a transformation of the cortex from a less mineralized structure to one that contains more highly mineralized bone. Closure of pores and increased mineralisation were both delayed in $Dmp1^{Cre}$:$Socs3^{f/f}$ mice, consistent with very early work showing that new bone deposited at the growth plate is progressively replaced with bone of higher mineral density as the bone ages (**Enlow, 1962**), and more recent work showing that adolescent bone has higher mineral content than infant bone (**Zimmermann et al., 2019**). Clearly in the $Dmp1^{Cre}$:$Socs3^{f/f}$ mice, this process is slowed, a defect that may explain the reduced ductility of bone in the presence of normal bone mass in the 6 month old female $Dmp1^{Cre}$:$Socs3^{f/f}$ mice (**Cho et al., 2017**). This raised the question of whether delayed cortical development in childhood could lead to compromised bone strength later in life. We found here that the alteration in cortical

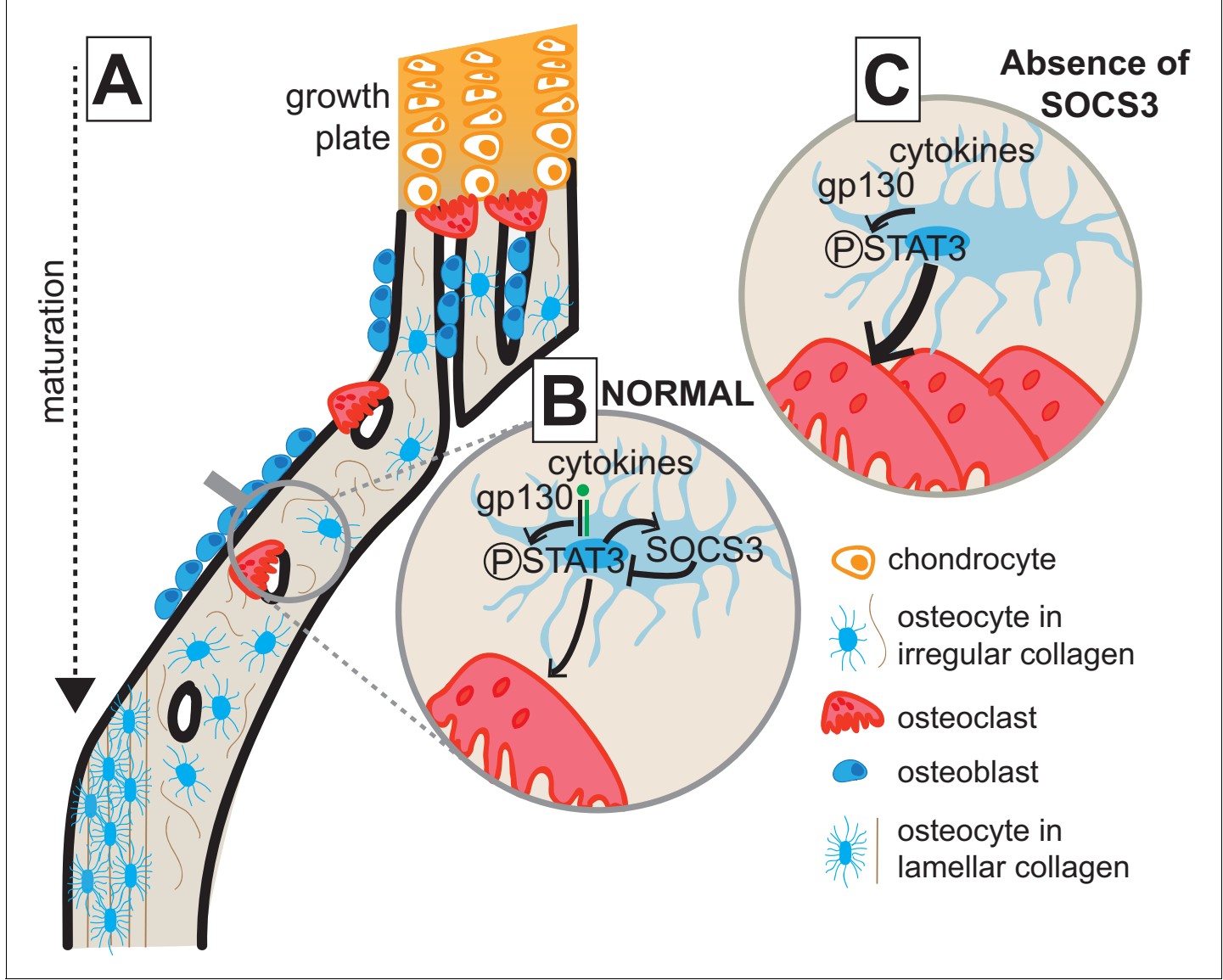

**Figure 7.** Model of cortical maturation and the role of SOCS3 in osteocytes. (A) During longitudinal growth, trabeculae arising from remodelling of the growth plate consolidate, initially forming porous woven cortical bone with irregular oriented osteocytes with rounded cell bodies. This bone is subsequently remodelled to form less porous, more highly mineralized lamellar cortical bone, with flattened regularly-oriented osteocytes. (B) During normal cortical bone formation IL-6 family cytokines signal through gp130 in osteocytes, initiating STAT3 phosphorylation. This stimulates osteoclast formation within cortical pores and on the endocortical surface; the STAT3 signal and subsequent osteoclast formation is moderated by the SOCS3 negative feedback loop. (C) In the absence of SOCS3 in osteocytes, the unchecked STAT3 phosphorylation is prolonged, leading to increased osteoclast formation, and increased remodelling of the cortical bone, delaying the formation of the more highly mineralized lamellar structure.

bone material behaviour generated by *Socs3* deficiency, like other aspects of the phenotype, was corrected by deletion of gp130.

Measurement of cortical porosity is a technique now common in micro-computed tomography studies of murine bone, but in this case, measuring the cortical pores in the metaphysis was challenging due to the difficulty of defining cortical and trabecular bone compartments. At the resolution of a standard micro-CT instrument, the pores measured by this technique are significantly larger than osteocyte lacunae, and the majority of these pores are 'open pores' that connect with other pores, and with the top or bottom of the measurement region. In control mice, very few pores were observed, and the smallest closed pores detected in the metaphyseal region were 5–10,000 micron (at the lower limit of detection for this parameter); this is approximately 10x the size of an osteocyte

lacuna (*Buenzli and Sims, 2015*). In 12 week old female *Dmp1*$^{Cre}$:*Socs3*$^{f/f}$ samples, the closed pores were much larger, on average 146,000 ± 33,000 micron. This is approximately the size of 150 osteocyte lacunae (in 3 dimensions). They are complex pores – with a surface area to volume ratio of (on average) 120 ± 24. This indicates they are large enough to contain blood vessels, osteoclasts, osteoblasts, and bone marrow, and similar to the pores detected histologically in *Figure 6F and G*.

We were surprised to observe a significant transformation from mid- to high-density bone in control mice between 12 and 15 weeks, after the cortical bone had formed. It is often surmised that rodent bone does not remodel after the cortical shape has been established. However, although resorption cavities have only been described in murine bone after 52 weeks of age, lamellar bone content in bone increases even between 22 and 91 weeks of age; this occurs through an asymmetrical process where woven bone on the periosteum or the endocortical surface is resorbed and replaced by lamellar bone (*Ferguson et al., 2003*). However, in the control mice, it is surprising that such a dramatic increase in the proportion of highly mineralized bone has occurred within a three week time period, suggesting that the increase in mineral content may also reflect secondary mineralization of the established bone matrix.

This transition to bone of higher material density, and the difficulty of defining cortical bone in the *Dmp1*$^{Cre}$:*Socs3*$^{f/f}$ model, led us to develop a new non-invasive method to quantify cortical bone maturation in the metaphysis, which may be applied to a range of biological and medical applications. This unbiased method also helped us to resolve underlying concerns about the subjective definitions of cortical and trabecular bone in mouse models with atypical bone structure. To achieve this, we used multi-level Otsu thresholding. This method makes use of an algorithm that segments pixels from the raw micro-CT images into different classes based on the gray level intensities within the image (*Otsu, 1979*). Even though the micro-CT scans did not exhibit deep troughs in pixel intensity, Otsu-based segmenting thresholds were highly consistent across samples. This provided a relatively simple way to measure the changes in bone maturation that occur with bone age in the same tissue samples, since the metaphysis contains both young and older bone. Furthermore, this method enabled comparisons between animal models and subjects in which cortical bone development is modified; this had previously been challenging due to the peculiar nature of the *Dmp1*$^{Cre}$:*Socs3*$^{f/f}$ bone and may be valuable for other models in which the distinction between trabecular and cortical bone is difficult to define, such as models of osteopetrosis, chronic kidney disease or vitamin D deficiency. Since the method is non-invasive, application to human CT scans or primate scans could provide additional information about changes in bone structure during growth and ageing or in response to pharmacological agents, including predicting fracture in patients with normal bone mass. Since it is non-destructive, it may also be useful for assessing bone in bioarchaeological, forensic, or other archived human or zoological specimens.

The delayed metaphyseal development of *Dmp1*$^{Cre}$:*Socs3*$^{f/f}$ bone indicates that cortical formation requires SOCS3 expression by *Dmp1*$^{Cre}$ positive cells (most likely late-stage ostoblasts and osteocytes). SOCS3 is an intracellular protein that provides negative feedback for a wide range of cytokine receptors, including leptin receptor, G-CSF, and all IL-6 family receptor subunits that signal by complexing with gp130 (*Morris et al., 2018*). Since the presence or absence of specific receptors in osteocytes is difficult to prove due to lack of purity of cell preparations, and the loss of the osteocyte phenotype when primary cells are cultured (*Chia et al., 2015*), we tested STAT3 responses to oncostatin M, leukaemia inhibitory factor, IL-11, leptin, and G-CSF, and how they were modified in *Dmp1*$^{Cre}$:*Socs3*$^{f/f}$ bone. Although pharmacological treatment with erythropoietin modifies bone mass, and activates JAK/STAT signalling, we did not test the response to erythropoietin since in vivo lineage tracing studies have shown that osteoblast lineage cells do not express this receptor (*Singbrant et al., 2011*). G-CSF did not induce a STAT3 phosphorylation response even in control animals, consistent with earlier studies reporting that G-CSFR mRNA (*Csf3r*) is not found in cultured osteoblast lineage cells (primary osteoblasts, osteoblast cell lines, and the MLO-Y4 osteocyte-like cell line) (*Katayama et al., 2006*). We also observed no STAT3 response to leptin, a cytokine hypothesized to have both direct and indirect effects on bone formation (*Takeda et al., 2002*; *Yue et al., 2016*; *Ducy et al., 2000*). In contrast, the gp130-dependent cytokines Oncostatin M (OSM), Leukaemia inhibitory factor (LIF), and IL-11 all induced STAT3 phosphorylation. OSM and LIF have been shown previously to stimulate STAT3 phosphorylation in osteoblasts cultured to express osteocyte markers in vitro, and to suppress sclerostin expression in osteocytes in vitro and in vivo (*Walker et al., 2010*; *Walker et al., 2016*). A response of osteocytes to IL-11 has not been reported

previously, but IL-11 stimulates STAT3 phosphorylation in primary calvarial osteoblasts (*Romas et al., 1996*), and we have observed downregulation of sclerostin by IL-11 in an osteosarcoma-derived cell line (UMR106-01, Patricia W.M. Ho, unpublished data). In *Dmp1^{Cre}:Socs3^{f/f}* bone, STAT3 responses to these cytokines were elevated and prolonged in mice lacking SOCS3. This indicates that at least three members of this cytokine family may require suppression to enable cortical maturation.

Indeed, when gp130 was deleted in osteocytes, in *Dmp1^{Cre}:Socs3^{f/f}:Il6st^{f/f}* mice, the *Dmp1^{Cre}: Socs3^{f/f}* phenotype – including the high cortical porosity and the high proportion of low density bone close to the growth plate – was prevented. This not only normalised the cortical porosity, and bone density maturation pattern, but also reduced osteocytic STAT3 phosphorylation and the high level of intracortical bone resorption of *Dmp1^{Cre}:Socs3^{f/f}* mice. This suggests gp130-dependent STAT3 phosphorylation in cortical osteocytes must be suppressed to limit osteoclast formation in the developing cortex, and that continued resorption at the periphery prevents trabecular coalescence.

While we identified that gp130 is the receptor that must be suppressed for cortical consolidation, we did not identify a responsible hyperactive gp130-dependent cytokine. There are many candidates since gp130 is a signalling receptor for a wide range of bone-active cytokines (*Sims, 2015*; *Sims, 2016*). Previously, we identified that IL-6 is not responsible for the *Dmp1^{Cre}:Socs3^{f/f}* phenotype since it was not improved by deleting IL-6 (*Cho et al., 2017*). Our western blot data suggest that IL-11, OSM and LIF are all candidates. All three showed extended STAT3 phosphorylation in the *Dmp1^{Cre}:Socs3^{f/f}* mice, and all three are required for normal osteoclast formation (*Sims et al., 2005*; *Walker et al., 2010*; *Poulton et al., 2012*). There are other candidates that we did not test: Cardiotrophin-1 may be involved since it modifies osteoclast formation and activity in vivo (*Walker et al., 2008*), and acts directly on osteocytes to reduce sclerostin expression (*Walker et al., 2010*); CNTFR-binding cytokines may also play a role, since CNTFR is expressed in osteocytes (*McGregor et al., 2010*), but CNTFR-binding cytokines do not stimulate osteoclastogenesis (*McGregor et al., 2010*) so this seems unlikely. We suggest that IL-11, OSM, and LIF at least must be suppressed in osteocytes to limit their production of factors, such as RANKL, that promote osteoclastogenesis in the developing cortical region, thereby allowing bone formation to dominate and consolidate the cortical structure.

The effects of the gp130-dependent cytokines on cortical consolidation appear to be local, since high levels of osteoclast formation were not observed in trabecular bone of female *Dmp1^{Cre}:Socs3^{f/f}* mice (*Cho et al., 2017*). This indicates separate mechanisms controlling osteoclast formation in the trabecular bone of the secondary spongiosa and the regions undergoing cortical consolidation of the metaphysis. Differential effects of cytokines and receptors in cortical and trabecular bone have been noted previously (*Johnson et al., 2014*; *Calvi et al., 2001*), although mechanisms to explain these differences remain elusive (*Sims and Vrahnas, 2014*). The cellular source and regulation of the critical cytokines may differ. This may relate to differences in mechanical strain between the regions: IL-11, OSM, OSM receptor, SOCS3 and STAT3 mRNA levels are all rapidly increased in bone in response to in vivo mechanical loading (*Mantila Roosa et al., 2011*), and mice lacking STAT3 in osteoblasts or in osteocytes both have impaired anabolic response to mechanical load (*Zhou et al., 2011*; *Corry et al., 2019*). Another possibility is differences in vascularisation and supply of osteoclast precursors between the sites, since LIF regulates vascular formation at the growth plate, but not in remodelling bone (*Poulton et al., 2012*). Our preliminary analysis of adult calvarial bone suggests that these mechanisms also exist in calvarial bone, even though bone formed at that site is generated through intramembranous, rather than endochondral, ossification. Further definition of actions of these cytokines in the developing cortical bone, both in the metaphysis and the calvariae, will require analysis of developing skeletons in cell-specific knockouts of these receptors.

In this study, we developed a new mouse model of gp130 deletion in osteocytes. Previously, we used mice generated with an earlier gp130-flox mouse model that targeted deletion of the transmembrane domain, rather than the first exon (*Betz et al., 1998*). Although that model blocked gp130-mediated intracellular responses, it also raised levels of soluble gp130 (sgp130) in target cells (*Betz et al., 1998*). Soluble gp130 (sgp130) circulates in the serum as at least 3 isoforms (full-length sgp130 – the dominant form, sgp130-RAPS and sgp130-E10), and has been detected in concentrations of 390 ng/ml in healthy adults (*Narazaki et al., 1993*). Endogenous sgp130 can exist as a monomer or a dimer (*Wolf et al., 2016*) and is generated by alternative splicing (*Narazaki et al.,*

*1993*; *Tanaka et al., 2000*; *Zhang et al., 1998*; *Sommer et al., 2014*), alternative intronic polyadenylation (*Sommer et al., 2014*), and may be released by proteolytic cleavage (*Montero-Julian et al., 1997*; *Sherwin et al., 2002*; *Zhou et al., 1998*). All three forms of sgp130 can inhibit interleukin 6 (IL-6) action; full length sgp130 also inhibits ciliary neurotrophic factor (CNTF), oncostatin M (OSM) and leukaemia inhibitory factor (LIF) (*Narazaki et al., 1993*). In our previous model, recombination of gp130 in osteocytes led to an impaired bone formation response to OSM, and impaired bone formation and reduced mechanical strength (*Johnson et al., 2014*). That phenotype may have resulted either from lack of gp130 signalling, or from inhibition due to elevated local production of soluble gp130, by osteocytes. To eliminate the possibility that the phenotype was due to elevated sgp130 production from osteocytes, we used a new gp130-flox mouse (*Il6st^{f/f}*) targeting exon 1, to delete all soluble isoforms. This resulted in a phenocopy of the earlier *Dmp1^{Cre}:gp130^{f/f}* mouse, confirming that the earlier phenotype was not caused by sgp130 production, but by deletion of gp130 signalling within the target cells, and that signalling through gp130 in osteocytes maintains normal trabecular bone formation, and limits cortical width.

Mature murine cortical bone does not establish the Haversian systems characteristic of larger mammals, including humans. However, humans do not yet exhibit Haversian systems in early stages of metaphyseal development; at that stage, this region is made up of thin, porous, non-coalesced bone (*Rauch, 2012*), similar to the structure observed in the mouse model. This is why the border with the growth plate is a site of greater bone fragility (*Rauch, 2012*). As the cortical bone becomes more compact it passes through a transitional state of 'coarse, irregular whorls or convolutions' of compact bone, described in Rhesus monkey by *Enlow, 1962*. This bone is only later remodelled to form Haversian systems. We observed this transitional 'coarse compact' bone form in the metaphysis close to the growth plate in both control and *Dmp1^{Cre}:Socs3^{f/f}* bone, and coarse compact (woven) bone was still seen in the *Dmp1^{Cre}:Socs3^{f/f}* metaphyseal region close to the diaphysis even at 12 weeks of age; it was seen particularly clearly with Ploton silver staining. In contrast, this region had been partially replaced with lamellar bone in the control animals. This suggests that the key difference in cortical development between small (non-Haversian) and large (Haversian) bone occurs after the transitional 'coarse compact' bone phase. The initial steps in cortical bone development are therefore likely to be the same in large and small mammals. Whether the STAT3/SOCS3 signalling is involved in this transition, independent of the establishment of Haversian systems, in larger mammals remains to be established.

In conclusion, this study shows (1) that the formation of consolidated cortical bone in the metaphysis involves both the closure of cortical pores by trabecular coalescence, and the replacement of low density bone with high density bone, (2) that these processes can be quantified, and (3) that this process requires suppression of gp130 cytokine signalling by SOCS3 in osteocytes.

## Materials and methods

### Key resources table

| Reagent type (species) or resource | Designation | Source or reference | Identifiers | Additional information |
|---|---|---|---|---|
| Strain, strain background (*Mus musculus*) | *Dmp1^{Cre}* | Lynda Bonewald (Indiana University, USA) | Tg(Dmp1-cre)1Jqfe | C57BL/6 background |
| Strain, strain background (*Mus musculus*) | Socs3^{tm1Wsa} | Warren Alexander (Walter and Eliza Hal Institute) | Socs3^{tm1Wsa} | C57BL/6 background |
| Strain, strain background (*Mus musculus*) | *Il6st-flox* mouse strain | NCRR-NIH-supported KOMP Repository | IL6st^{tm1a(KOMP)Mbp} | C57BL/6 background |
| Antibody | phospho-STAT3 (Rabbit polyclonal) | Cell Signalling | Cat #9131 RRID:AB_331586 | IHC (1:100), WB (1:400) |
| Antibody | STAT3 (mouse monoclonal) | Cell Signalling | Cat#9139 RRID:AB_331757 | WB (1:400) |

*Continued on next page*

Continued

| Reagent type (species) or resource | Designation | Source or reference | Identifiers | Additional information |
|---|---|---|---|---|
| Antibody | Goat anti-Mouse | Dako | Cat#P0447 RRID:AB_2617137 | WB (1:1500) |
| Antibody | Swine anti-Rabbit | Dako | Cat#P0217 RRID:AB_2728719 | IHC (1:250) WB (1:2000) |
| Peptide, recombinant protein | Murine oncostatin M | R and D Systems | Cat#495-MO-025 | |
| Peptide, recombinant protein | human IL-11 | R and D Systems | Cat#218-IL-025 | |
| Peptide, recombinant protein | murine LIF | Merck Millipore | #LIF2010 | |
| Peptide, recombinant protein | Murine Leptin | R and D Systems | #498-OB-01M | |
| Peptide, recombinant protein | murine G-CSF | R and D Systems | #414-CS-005 | |

## Mouse strains and tissue collection

*Dmp1^Cre^:Socs3^f/f^* mice were generated as previously described (*Cho et al., 2017*) by crossing *Dmp1^Cre^* mice (Tg(Dmp1-cre)1Jqfe), provided by Lynda Bonewald (Indiana University, USA), with SOCS3 floxed mice (Socs3^tm1Wsa^), provided by Warren Alexander (Walter and Eliza Hall Institute of Medical Research, Melbourne, Australia) (*Croker et al., 2003*). Both were backcrossed to a C57BL/6 background. *Dmp1^Cre^* genotypes were confirmed with the following primers provided by Carl Walkley, St. Vincent's Institute: forward 5'-GAC CAG GTT CGT TCA CTC ATG G-3', reverse 5'-AGG CTA AGT GCC TTC TCT ACA-3'. *Socs3* recombination was detected with the following primers: forward 5'-GAG TTT TCT CTG GGC GTC CTC CTA-3', reverse 5'- TGG TAC TCG CTT TTG GAG CTG AA-3'; all experimental mice were genotyped. Successful knockdown of *Socs3* in bones from *Dmp1Cre: Socs3^f/f^* mice has been reported previously (*Cho et al., 2017*).

The *Il6st-flox* mouse strain (IL6st^tm1a(KOMP)Mbp^) was generated from an ES cell clone (DEPD00539_7_F12) obtained from the NCRR-NIH-supported KOMP Repository. The mutation in the *IL6st* gene was generated using knockout-first allele technology (*Testa et al., 2004*). Heterozygous IL6st ^tm1a(KOMP)Mbp^ mice were generated by the Australian Phenomics Network and were backcrossed onto C57BL6 background, and bred with FLP mice (kindly provided by Dr Carl Walkley, St Vincent's Institute) to induce FRT recombination and deletion of the LacZ/Neo cassette, and obtain mice carrying the conditional floxed allele *IL6st^f/f^*. Recombination was confirmed by genotyping with the following primers: P1 5'-GTC CCA CCA TCC TAA CCT CC-3', P2 5'- CAC GGT TCC AAA AGT TGA CC-3', P3 5'- CAA CGG GTT CTT CTG TTA GTC C-3' (IDT DNA). The FLP allele was bred out and mice were maintained on a C57BL/6 background. *IL6st^f/f^* mice were crossed with *Dmp1^Cre^* mice to generate *Dmp1^Cre^:Il6st^f/f^* mice. These were crossed with *Dmp1^Cre^:Socs3^f/f^* mice as required. All animal procedures were conducted with approval of the St. Vincent's Health Melbourne Animal Ethics Committee.

Bone and blood samples were collected from the above mouse lines at 12 and 15 weeks of age, after injection with calcein at 3 and 10 days prior to tissue collection (*Sims et al., 2000*). Sample sizes were based on previous studies using micro-CT and histomorphometry; no explicit power analysis was used. Mice were fasted for 12 hr prior to anaesthesia with ketamine/xylazine and blood collection via cardiac puncture. Flushed femora were harvested for RNA extraction as previously described (*Walker et al., 2012*). cDNA synthesis and quantitative PCR were performed as previously described (*McGregor et al., 2019*). Briefly, AffinityScript (Agilent Technologies 600559) was used to generate cDNA, and qPCR was performed using Brilliant II SYBR Green Mastermix (Agilent Technologies 600828) with cycling conditions as follows: 10 mins at 95°C followed by (95°C for 30 s and 60°C for 1

min) for 40 cycles followed by a dissociation step (95°C for 1 min, 55°C for 30 s and 95°C for 30 s); primers for *Il6st* (*Walker et al., 2016*), *Tnfsf11* (*Allan et al., 2008*), *Tnfsfr11b* (*Nakamura et al., 2007*), *Acp5* (*Chia et al., 2015*), *Hprt1* (*Gooi et al., 2010*) and *B2m* (*Winkler et al., 2005*) were described previously.

## Micro-computed tomography

Female *Dmp1$^{Cre}$* and *Dmp1$^{Cre}$:Socs3$^{f/f}$* mice were scanned at 12, 13, 14 and 15 weeks of age. For each scan, mice were anaesthetised with a single i.p. injection of (10% ketamine, 10% xylazil, 80% saline in 10 µL per gram body weight). Once anaesthetised, their legs were immobilised with a polystyrene cut-out to straighten the femur before scanning. The x-ray source was set at 44kV and 220µA. Projections were acquired over 180° (step of 0.6°), 9 µm voxel resolution, 0.5 mm aluminium filter, 2300 ms exposure time, frames averaging 1. Image slices were reconstructed and analysed using NRecon V1.7.0.4, Dataviewer V1.5.2.4 and CT Analyser V1.16.4.1. 3D- tomographic volume images of bone were created using CTVox V3.2.0. At 16 weeks of age, mice were culled and femora fixed in 4% paraformaldehyde and scanned under the same conditions. A second cohort of age-, sex- and genotype-matched mice were scanned at 12 and 15 weeks of age to test whether repeated anaesthesia and radio-exposure (*Klinck et al., 2008*; *Laperre et al., 2011*; *Sacco et al., 2017*) modified bone mass. Fixed bones from male and female *Dmp1$^{Cre}$*, *Dmp1$^{Cre}$:Il6st$^{f/f}$* and *Dmp1$^{Cre}$:Socs3$^{f/f}$:Il6st$^{f/f}$* mice were scanned after fixation, under the same conditions.

After scanning, femoral length was measured in all samples collected. Trabecular bone mass was assessed in the metaphysis, in a region of interest that was 15% the length of the bone, commencing at a distance 7.5% of bone length from the distal femoral growth plate. To determine whether bone loss occurred by the repeated scanning, global thresholds for analysis of trabecular bone were set at a minimum of 0.238 g/cm$^3$ and cortical bone at a minimum of 0.632 g/cm$^3$ calcium hydroxyapatite. No significant difference was detected in trabecular or cortical bone mass between the mice scanned weekly and those scanned only at 12 and 15 weeks, confirming that the repeated in vivo scanning did not cause bone loss (data not shown).

To visualise changes in cortical consolidation over time, pseudocolorized images were used to create videos with CTVox V3.2.0 'Flight Recorder'. Images for each mouse from each consecutive scan were merged after geometrically aligning with 12-week-old images using Dataviewer v1.5.2.4.

Since the flight recorder images showed changes in mineral density over time, even in wild type mice, we developed a method to quantify newly forming cortical bone at three different levels of mineral density. To measure bone areas of low, medium, and high density bone, we used nonparametric unsupervised 4 level Otsu thresholding using CT Analyser V1.16.4.1. Otsu thresholding is a thresholding algorithm that segments the pixels into different classes based on the gray level intensities within the image (*Otsu, 1979*). We used 4 level Otsu thresholding across the entire metaphyseal region of 15 week old *Dmp1$^{Cre}$* femora to class the gray pixels into four density levels representative of the 4 densities within the mature metaphysis from 15 week old wild type bone. The threshold ranges for each of these levels were calibrated to calcium hydroxyapatite standards, as follows: low density (0.632–1.143 g/cm$^3$ CaHA); mid density (1.143–1.528 mg/cm$^3$ CaHA) and high density (>1.528 g/cm$^3$ CaHA). The lowest density quartile (0–0.632 g/cm$^3$ CaHA) was excluded; although it contained some low-density bone, it also contained low density material detected outside the bone, suggesting inclusion of noise. Representative images showing three levels used and the matched raw image at the distal metaphysis are shown in *Figure 2A*. Within the 15 week old *Dmp1$^{Cre}$* samples, despite the lack of deep troughs in pixel intensity, the segmenting thresholds were highly consistent across the group of samples; with the lowest threshold having a range of grayscale values of 76–85 (11% variance), mid-density threshold ranging from 130 to 137 (5% variance), and high-density ranging from 170 to 176 (3.5% variance). Samples from both genotypes and ages were subjected to analysis at the same three density thresholds. Bone volume at each of these densities were measured and are expressed relative to the total cross sectional volume. In addition, to obtain information about cortical bone maturation throughout the metaphysis, data were extracted using CT Analyser V1.16.4.1 for each 9 micron slice from the proximal to distal edge of the metaphyseal region, and bone areas at each density were calculated as a percentage of total cross sectional area for each slice (*Figure 3A*).

Cortical porosity and bone volume at each of the three densities were measured in female *Dmp1$^{Cre}$:Socs3$^{f/f}$* and littermate *Dmp1$^{Cre}$* (control) mice at 12 and 15 weeks of age (n = 9–10/group),

and in male and female 12 week old $Dmp1^{Cre}$:$Socs3^{f/f}$ and $Dmp1^{Cre}$:$Socs3^{f/f}$:$Il6st^{f/f}$ mice. These were measured in the metaphysis (as described above). At 12 weeks of age, cortical porosity was also measured in the diaphysis (in a region of interest that was 15% the length of the bone, commencing at a distance 30% of bone length from the distal femoral growth plate). Cortical porosity measurements were determined with bone defined using a global threshold of a minimum of 0.238 g/cm$^3$ calcium hydroxyapatite. For the analysis of archived scans of 26 week old $Dmp1^{Cre}$:$Socs3^{f/f}$ and littermate $Dmp1^{Cre}$ (control) femora, since scans were performed under different conditions, segmentation levels were based on concurrent scans of 26 week old male control femora. These were: low density (0.565–1.036 g/cm$^3$ CaHA); mid density (1.036–1.432 mg/cm$^3$ CaHA) and high density (>1.432 g/cm$^3$ CaHA).

Mechanical testing of femora from 12 week old $Dmp1^{Cre}$:$Socs3^{f/f}$ and $Dmp1^{Cre}$:$Socs3^{f/f}$:$Il6st^{f/f}$ mice was performed by three-point bending tests as previously described (McGregor et al., 2019). Two samples were excluded on the basis of abnormal force/displacement curves, although data fell within the normal range. All micro-CT measurements and mechanical testing were conducted with the observer being blinded to the genotype of the samples.

## STAT3 phosphorylation responses to in vivo cytokine treatments

To determine how SOCS3-dependent cytokine signalling was modified in $Dmp1^{Cre}$:$Socs3^{f/f}$ mice, 6 week old male $Dmp1^{Cre}$ and $Dmp1^{Cre}$:$Socs3^{f/f}$ mice (n = 4/group) were anaesthetised using Isoflurane inhalant (FORTHANE, AbbVie) and injected with a single subcutaneous dose of Vehicle (PBS (Sigma # D8537) +2% heat-inactivated mouse serum), murine Oncostatin M (R and D Systems #495-MO-025), human IL-11 (R and D Systems 218-IL-025#), murine LIF (Merck Millipore #LIF2010), murine Leptin (R and D Systems #498-OB-01M), or murine G-CSF (R and D Systems #414-CS-005) each at a final concentration of 0.2 µg in a volume of 25 µl as previously described (Walker et al., 2008; Cornish et al., 1993). At 0', 30', 1 hr, 2 hr, 4 hr and 6 hr mice were sacrificed by cervical dislocation, the calvarium harvested and quickly snap frozen in liquid nitrogen. Once all animals had been collected, the calvariae were dissected in liquid nitrogen into smaller fragments. Fragments were placed directly in a 1.5 ml NAVY bead Lysis Kit (Biotools, #NavyE1-RNA) in 0.2 ml RIPA modified buffer (150 mM NaCl, 1 mM EDTA, 1% Triton-X 100, 1% sodium deoxycholate, 0.1% SDS, 20 mM Tris) and protein was extracted via mechanical disruption using the Bullet Blender Storm24 (5 min, speed 12). The supernatant was spun at 13,000 rpm (Eppendorf 5424 (24 × 3,75 g)) for 5 min at 4°C, and whole cell lysate collected. Protein concentration was determined by BCA (Pierce #) as per the manufacturer's instructions. 30 ug protein was loaded onto a 4–12% Bis-Tris gel (Thermo Fisher) and separated under reducing conditions as per manufacturer's instructions. Proteins were transferred to PVDF membrane (Roche #03010040001) using wet transfer method (Xcell II Invitrogen EI9051). Each membrane was probed with either p-STAT3 Y705 (Cell Signalling #9131), or STAT3 (124H6) (Cell Signalling #9139), each at 1:400 in 5% skim milk/TBST. Secondary antibodies used were either Goat anti-Mouse (Dako #P0447) at 1:1500 or Swine anti-Rabbit at 1:2000 (Dako #P0217) both in 5% skim milk/TBST. ECL Prime (Amersham #GEHERPN2232) and film (Fujifilm #497690) were used for the detection system.

## Histology, immunohistochemistry and histomorphometry

Femora and tibiae from 12 week old and 16 week old $Dmp1^{Cre}$, $Dmp1^{Cre}$:$Socs3^{f/f}$ and $Dmp1^{Cre}$:$Socs3^{f/f}$:$Il6st^{f/f}$ mice were decalcified in EDTA and embedded in paraffin as previously described (Sims et al., 1997). To visualise calcein labels, tibiae and calvarial samples were infiltrated and embedded in methylmethacrylate as previously described (Sims et al., 2006). Ploton silver stain and TRAP stains were carried out on paraffin embedded decalcified samples, or in methylmethacrylate coronal sections for calvarial samples, as previously described (Jáuregui et al., 2016; Sims et al., 2004). Void areas within the calvarial bone were measured along the calvarial bone, excluding the central 740 microns including the parietal suture, as previously described (McGregor et al., 2020). Immunohistochemistry for phospho-STAT3 was carried out by the following method. 5 µm sections were dewaxed in Histoclear, rehydrated through graded ethanols and endogenous peroxidase blocked for 30 min in 100% methanol. Samples were permeabilized for 15 min in 0.5% trypsin in PBS, and rinsed. Before primary antibody was applied, samples were blocked with 10% swine serum in PBS with 0.01% Tween 20 (TBS) for 60 min. Phosph-STAT3 antibody (Cell Signalling #9131, 1:100),

or IgG control (rabbit IgG, Dako) was applied in a humidified chamber at room temperature overnight. After rinsing, a secondary antibody (swine anti-rabbit, Dako) was applied for 45 min in TBS at 1:250, followed by streptavidin horseradish peroxidase (Dako) 1:500 in TBS for 45 min then biotin tyramine 1:50 in amplification diluent for 7 min (TSA Biotin system kit, Perkin Elmer, Waltham, USA). Samples were rinsed in TBS containing Triton 1000 between each step. Detection was achieved using a diaminobenzidine colorimetric kit (Dako) and counterstained with Mayer's hematoxylin. Histomorphometric measurements of phospho-STAT3 positive osteocytes, osteoclast surface, osteoblast surface, and void areas were all carried out in cortical bone of the lateral tibial metaphysis, commencing 370 µm below the distal end of the growth plate, and extending for 1.11 mm. All histomorphometric measurements were conducted with the observer being blinded to the genotype of the samples.

## Statistical analyses

Statistical analyses were carried out using GraphPad Prism 8. In most instances, two-way ANOVA with mixed-effects analysis was used, with repeated measures for slice-by-slice comparisons. Sidak post hoc tests were used to identify significant differences. A $p < 0.05$ was considered significant. No outliers were excluded from any analyses.

# Acknowledgements

The authors thank the staff of the St. Vincent's Health Bioresources Centre for excellent animal care and assistance, and Brett Tonkin, Holly Brennan, Blessing Crimeen-Irwin, and Daniel Whelan for technical assistance with micro-CT, and histomorphometry. This work was supported by NHMRC Grants 1120978 and 1058625 to NAS and TJM. NAS was supported by an NHMRC Senior Research Fellowship and by the SVI Brenda Shanahan Fellowship. St Vincent's Institute acknowledges the support of the Victorian State Government OIS program.

# Additional information

### Funding

| Funder | Grant reference number | Author |
| --- | --- | --- |
| National Health and Medical Research Council | 1120978 | Natalie A Sims |
| National Health and Medical Research Council | 1154819 | Natalie A Sims |
| National Health and Medical Research Council | Senior Research Fellowship | Natalie A Sims |
| St. Vincent's Institute | Brenda Shanahan Fellowship | Natalie A Sims |

The funders had no role in study design, data collection and interpretation, or the decision to submit the work for publication.

### Author contributions

Emma C Walker, Formal analysis, Investigation, Visualization, Writing - review and editing; Kim Truong, Investigation, Visualization; Narelle E McGregor, Formal analysis, Investigation, Visualization, Methodology, Writing - review and editing; Ingrid J Poulton, Data curation, Investigation, Visualization, Methodology; Tsuyoshi Isojima, Formal analysis, Investigation; Jonathan H Gooi, Formal analysis, Investigation, Methodology, Writing - review and editing; T John Martin, Conceptualization, Funding acquisition, Writing - review and editing; Natalie A Sims, Conceptualization, Data curation, Formal analysis, Supervision, Funding acquisition, Investigation, Methodology, Writing - original draft, Project administration, Writing - review and editing

### Author ORCIDs

Natalie A Sims  https://orcid.org/0000-0003-1421-8468

### Ethics

Animal experimentation: The study was performed with the recommendations for the Australian code for the care and use of animals for scientific purposes of the National Health and Medical Research Council (Australia). All animals were handled according to protocols approved by the Animal Ethics Committee of St. Vincent's Health Australia (Melbourne) (approval 026/15).

### Decision letter and Author response

Decision letter https://doi.org/10.7554/eLife.56666.sa1
Author response https://doi.org/10.7554/eLife.56666.sa2

## Additional files

### Supplementary files

- Transparent reporting form

### Data availability

All data generated or analysed during this study are included in the manuscript and supporting files.

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
