## [Decision Letter]

**Acceptance summary:**

Your study nicely focuses on a specified role of gp130 signaling in the regulation of cortical bone mass. Not much is known about how cortical bone is remodeled and lost, and in particular how cortical porosity effects bone strength. This study uniquely highlights the role of closure of cortical pores in determining bone strength and its mechanism. Cutting-edge methods that are widely applicable have been utilized, and you have provided a thoughtful and thorough response to the review critique.

**Decision letter after peer review:**

Thank you for submitting your article "Cortical bone maturation in mice requires SOCS3 suppression of gp130/STAT3 signalling in osteocytes" for consideration by *eLife*. Your article has been reviewed by two peer reviewers, and the evaluation has been overseen by Mone Zaidi as the Reviewing Editor and Clifford Rosen as the Senior Editor. The following individual involved in review of your submission has agreed to reveal their identity: Ulf Lerner (Reviewer #1).

The reviewers have discussed the reviews with one another and the Reviewing Editor has drafted this decision to help you prepare a revised submission.

As the editors have judged that your manuscript is of interest, but as described below that additional experiments are required before it is published, we would like to draw your attention to changes in our revision policy that we have made in response to COVID-19 (https://elifesciences.org/articles/57162). First, because many researchers have temporarily lost access to the labs, we will give authors as much time as they need to submit revised manuscripts.

Summary:

The reviewers find the study interesting and a significant addition to our current knowledge on corticalization. Specifically, clear evidence is provided that deletion of gp130 specifically in osteocytes prolongs cytokine-induced STAT3 activation The reviewers also find methodological novelty in the development of an objective method based on unbiased multi-level Otsu thresholding of the CT-scans to determine how immature woven trabecular/cortical bone transforms into mature, well organized, cortical lamellar bone. They point out that these studies may have therapeutic implications in providing the framework for strategies that could increase cortical strength. Overall studies were well done and technically sound. With that said, the reviewers concurred that revision is required with the possible addition of new data to further strengthen the manuscript.

Essential revisions:

1) It would be functionally important to conduct mechanical testing to determine bone strength and quality in the respective genotypes and its correlation with the described phenotypes. At least some basic biomechanical tests on bone specimens that may already be available is considered essential.

2) To provide further mechanistic insights, the authors are encouraged to consider measuring local and circulating levels of CTX as well as RANKL (IHC and ELISA).

3) Figure 4: please discuss why basal level of pSTAT3 in genetically manipulated mice is virtually indistinguishable from controls (at least in panels A and C). It would also be informative to determine if cells lacking SOCS3 are sensitized to IL-6 family cytokines.

4) There is lack of clarity on what the pores shown in Figure 1C represent, specifically in relation to their size. Furthermore, do the pores shown in the histological sections in Figure 5F and G contribute to this overall porosity? Is the increased porosity (Figure 1C) noted in both metaphyseal and diaphyseal areas?

5) Figure 2B shows that bone volume of the metaphyseal area in *Dmp1^Cre^*:*Socc^3f/f^* mice is increased at 12 and 15 weeks, but one of the reviewers cannot find any statement on whether the mice were male or female. The reviewer considers it important to compare these data to those in the previous Nature Communication report. There is also the question whether the increased bone volume is solely due to delayed corticalization, or are there other contributory mechanisms?

6) The authors nicely demonstrate in Figure 4 that deletion of Socs3 in *Dmp1* expressing cells results in prolonged activation of pSTAT3 in calvarial bones after local injections of LIF, OSM or IL-11 and that such prolonged activation is not observed after G-CSF or leptin injections. Since EPO also activates STAT3 and has been shown to affect bone mass it would have been important to inject this cytokine.

7) Please provide data on an absent bone resorption response (in Table 1) as it relates to the statement in the subsection “Development of a new mouse with *Dmp1^Cre^*-targeted deletion of gp130”.

8) It is apparent that gp130 activation of STAT3 in *Dmp1* expressing cells needs to be dampened by SOCS3, but it is puzzling that deletion of this mechanism in *Dmp1* expressing late osteoblasts/osteocytes only affects osteoclast formation in metaphyseal pores during corticalization (gp130 cytokines are not likely to be expressed only in this area). One would anticipate that osteoclastogenesis in other parts of the skeleton also would be affected. It is argued in the Discussion section that osteoclast numbers in general are not affected by referring to the Nature Communication paper, where it is shown that osteoclast numbers in the secondary spongiosa are not affected by deletion of Socs3 in osteocytes. This might be due to low numbers of osteocytes in the trabecular bone in this area. The specific effect of SOCS3 during corticalization would be more convincing if the authors could count osteoclasts in other areas with higher numbers of osteocytes.

---

## [Author Response]

Essential revisions:1) It would be functionally important to conduct mechanical testing to determine bone strength and quality in the respective genotypes and its correlation with the described phenotypes. At least some basic biomechanical tests on bone specimens that may already be available is considered essential.

Thank you for this suggestion. We have now carried out three point bending tests on the *Dmp1^Cre^:Socs^3f/f^* and *Dmp1^Cre^*:*Socs^3f/f^:Il6st^f/f^* mice. To our delight, this has shown that the deletion of *Il6st* increases the post-yield deformation of the bones, the reverse of phenotype we previously reported when we compared the *Dmp1^Cre^:Socs^3f/f^* bones to controls. This provides additional evidence of a rescue, and reinforces the importance of normal cortical development to bone strength. This data is included in a new Table 3, described in the subsection “Rescue of the *Dmp1^Cre^:Socs^3f/f^* phenotype by *Dmp1^Cre^* -targeted deletion of gp130”, and discussed in the third paragraph of the Discussion section.

2) To provide further mechanistic insights, the authors are encouraged to consider measuring local and circulating levels of CTX as well as RANKL (IHC and ELISA).

In allocating tissues and samples in experiments we considered this but felt that measuring systemic levels of resorption markers would be unlikely to help us identify a mechanism for a local control mechanism since they address cumulative effects throughout the skeleton. As a way of addressing this, we have assessed RANKL (*Tnfsf11*), OPG (*Tnfrsf11b*), and TRAP (*Acp5*) mRNA levels in the flushed femoral samples (i.e. femoral cortex lacking the primary spongiosa). In this analysis, we see a mild increase in *Tnfrsf11b* and reduction in *Acp5* in the *Dmp1^Cre^:Socs^3f/f^:Il6st^f/f^* bones compared to the *Dmp1^Cre^:Socs^3f/f^* bones. This confirms our histomorphometric data, and suggests that the rescue may involve a local increase in OPG production rather than a reduction in RANKL. This data is now included in a new Table 2, described in the subsection “Rescue of the *Dmp1^Cre^:Socs^3f/f^* phenotype by *Dmp1^Cre^* -targeted deletion of gp130”.

3) Figure 4: please discuss why basal level of pSTAT3 in genetically manipulated mice is virtually indistinguishable from controls (at least in panels A and C). It would also be informative to determine if cells lacking SOCS3 are sensitized to IL-6 family cytokines.

The lack of difference in basal levels of pSTAT3 by Western blotting is consistent with observations in mice with SOCS3 deletion targeted to other cell types. In those studies, basal levels of pSTAT3 were not differentiated by Western blot (Croker et al., Journal of Immunology, 2003; Croker et al., 2004); this may be because of the low level of STAT3 phosphorylation in basal conditions relative to the stimulated response, or because the tissue contains a mixture of cell types, including cells with normal levels of SOCS3. Many studies have shown that a wide range primary cells lacking SOCS3 are sensitized to IL-6 family cytokines. Since cultured primary osteocytes lose their phenotype in vitro (see Chia et al., 2015), we have now measured pSTAT3 in vivo in *Dmp1^Cre^:Socs^3f/f^* osteocytes. We have added this data to Figure 4D and describe it in the subsection “STAT3 phosphorylation in response to IL-6 family cytokines is prolonged in *Dmp1^Cre^:Socs^3f/f^*” Bone, with reference to the above papers.

4) There is lack of clarity on what the pores shown in Figure 1C represent, specifically in relation to their size. Furthermore, do the pores shown in the histological sections in Figure 5F and G contribute to this overall porosity? Is the increased porosity (Figure 1C) noted in both metaphyseal and diaphyseal areas?

To clarify, the pores measured by micro-computed tomography are large pores, and are certainly larger than osteocyte lacunae (our resolution is insufficient for that); the pores shown in the histological sections in Figure 6F and G would certainly contribute to this porosity. The actual size of the pores is difficult to measure; they are measured in 3 dimensions, and 90% of them are open pores (open to the top or bottom of the metaphyseal region, through connections with other pores in the network), this means it is not possible to calculate their total size. The closed pores (fully encapsulated in bone) can be quantified. Most control samples did not exhibit measurable closed cortical pores; in those few that did, the pores were 5-10,000 micron^3^ (which was at the lower limit of detection for this parameter); this is approximately 10x the size of an osteocyte lacuna. In 12 week old female *Dmp1^Cre^:Socs^3f/f^* samples, the closed pores are on average 146,000 ± 33,000 micron^3^. This is approximately the size of 150 osteocyte lacunae (in 3 dimensions), so large enough to contain blood vessels, osteoclasts, osteoblasts, and bone marrow; they are very complex pores – with a surface area to volume ratio of (on average) 120 ± 24. Our goal in this study was to move away from the standard measure of cortical porosity precisely because it is difficult to understand its meaning in this mouse model. We have added more detail of the above into the Discussion section (fourth paragraph), and to the Materials and methods (subsection “Micro-computed tomography”).

To determine whether the increased porosity also exists in the diaphysis, we have now assessed diaphyseal cortical porosity in 12 week old female *Dmp1^Cre^:Socs^3f/f^* mice; indeed it was significantly elevated. This data is included in the new Figure 1—figure supplement 1, and described in the Results section (subsection “Visualisation of cortical maturation between 12 and 15 weeks of age in murine femora and its delay in *Dmp1^Cre^:Socs^3f/f^*mice”).

5) Figure 2B shows that bone volume of the metaphyseal area in Dmp1^Cre^:Socs^3f/f^ mice is increased at 12 and 15 weeks, but one of the reviewers cannot find any statement on whether the mice were male or female. The reviewer considers it important to compare these data to those in the previous Nature Communication report. There is also the question whether the increased bone volume is solely due to delayed corticalization, or are there other contributory mechanisms?

The analysis shown in Figure 2B is of female mice. The sex of mice was mentioned in the Materials and methods section and has now been added to the figure legend for clarity. The greater total bone volume in female *Dmp1^Cre^:Socs^3f/f^* mice observed in the present study is consistent with our original observation of increased “trabecular bone volume” in female mice in our earlier Nature Communications paper; we have now added a note directly comparing this data to the earlier study in the Discussion (second paragraph).

Whether the delayed corticalisation is the sole cause of the high bone volume, or the high bone volume causes delayed corticalisation is not possible to determine.

6) The authors nicely demonstrate in Figure 4 that deletion of Socs3 in Dmp1 expressing cells results in prolonged activation of pSTAT3 in calvarial bones after local injections of LIF, OSM or IL-11 and that such prolonged activation is not observed after G-CSF or leptin injections. Since EPO also activates STAT3 and has been shown to affect bone mass it would have been important to inject this cytokine.

Although EPO activates STAT3 in bone, we have previously reported, using in vivo lineage tracing, that the EPO receptor is not expressed in cells of the osteoblast lineage (see Singbrandt et al., 2011); Furthermore, our unpublished RNAseq analysis of the Ocy454 murine osteocyte cell line detected only negligible levels of *Epor* RNA (reads of 0-5 counts, in comparison to LIFR which had reads of 800-1000 counts). Since testing each cytokine requires 44 age- and sex-matched mice per genotype all bred to be tested at the same time, we felt this would not be a justifiable experiment. We have added a comment about our rationale for not testing erythropoietin in the Discussion (seventh paragraph).

7) Please provide data on an absent bone resorption response (in Table 1) as it relates to the statement in the subsection “Development of a new mouse with Dmp1^Cre^-targeted deletion of gp130”.

Apologies for this omission – it has been added to Table 1.

8) It is apparent that gp130 activation of STAT3 in Dmp1 expressing cells needs to be dampened by SOCS3, but it is puzzling that deletion of this mechanism in Dmp1 expressing late osteoblasts/osteocytes only affects osteoclast formation in metaphyseal pores during corticalization (gp130 cytokines are not likely to be expressed only in this area). One would anticipate that osteoclastogenesis in other parts of the skeleton also would be affected. It is argued in the Discussion section that osteoclast numbers in general are not affected by referring to the Nature Communication paper, where it is shown that osteoclast numbers in the secondary spongiosa are not affected by deletion of Socs3 in osteocytes. This might be due to low numbers of osteocytes in the trabecular bone in this area. The specific effect of SOCS3 during corticalization would be more convincing if the authors could count osteoclasts in other areas with higher numbers of osteocytes.

To assess whether osteoclast formation is increased at a second cortical site, we have returned to our archived samples of calvariae from 12 week old *Dmp1^Cre^:Socs^3f/f^* mice collected in the original study. The calvarial bone from these mice contains significantly more marrow space – termed diplöe at this location – than control. These holes contain both a higher level of calcein label and more osteoclasts, indicating a greater level of remodelling of cortical bone at this site. This is particularly interesting because calvarial bone does not undergo endochondral ossification. This provides further evidence of a specific stimulatory effect on osteoclast activity in cortical bone in *Dmp1^Cre^:Socs^3f/f^*mice. This data has been added to Figure 6—figure supplement 1B-D, and is described in the Results section (subsection “Rescue of the *Dmp1^Cre^:Socs^3f/f^* phenotype by *Dmp1^Cre^* -targeted deletion of gp130”) and Discussion.